# A tactile discrimination task to study neuronal dynamics in freely-moving mice

Filippo Heimburg [1,4], Nadin Mari Saluti [1,4], Josephine Timm [1,3,4], Avi Adlakha [2], Maria Helena Bortolozzo-Gleich [1], Jesús Martín-Cortecero[1], Melina Castelanelli[1], Matthias Klumpp [1], Lee Embray[1], Martin Both [1], Thomas Kuner[2] & Alexander Groh [1] ✉

Sensory discrimination tasks are valuable tools to study neuronal mechanisms of perception and learning, yet most rodent paradigms rely on head fixation. Here, we present a whisker-dependent go/no-go discrimination task for freely moving mice, compatible with high-resolution electrophysiology and calcium imaging. Adult male mice rapidly learned to discriminate aperture widths while foraging on a linear platform, enabling investigations of tactile thresholds, rule reversals, and behavioral flexibility. Neural recordings revealed distributed tactile coding across the thalamocortical system, with units tuned to both sensory and motor features, including whisking, head angle, and spatial position. Aperture selectivity emerged in the barrel cortex during learning, and cortical lesions impaired performance, highlighting cortical involvement in learning and task execution. The setup is modular, automated, and supports simultaneous recordings and imaging aligned to naturalistic behavior. This platform provides a powerful tool to dissect sensory processing and learning in ethologically relevant conditions.

The rodent whisker system serves as a prevalent model for studying somatosensation in the mammalian brain, providing fundamental insights into the mechanisms of sensory perception, learning, and neuronal coding at the single neuron and synaptic levels[1–12]. Since the discovery of the whisker-to-cortex system[13], increasingly advanced experimental approaches have been devised to elucidate the biology and behavioral function of whiskers, while remaining compatible with high-resolution neurophysiological recordings. The most advanced current state-of-the-art approaches employ recordings in head-fixed mice or rats, while they learn to report or discriminate sensory stimuli[6,14–19]. However, while the head-fixed approach presents the best current compromise between experimental control and biological relevance, head-fixed paradigms impose a limitation in accounting for a core ethological function of the whiskers: navigating mazes to search for food, which requires free head movements.

To overcome this limitation, we developed a go/no-go aperture discrimination task for mice inspired by earlier studies on rats[20,21], and combined the task with state-of-the-art single-unit electrophysiological recordings and miniscope imaging. We made modifications to the original design to accommodate rapid learning of multiple rules within an experimentally accessible timeframe in mice. The paradigm enables behavior that more closely resembles natural conditions (foraging in a freely moving apparatus) while allowing monitoring of decision-making processes, including licking, head movements, and whisker behavior. We show that neuronal activity can be monitored over the course of several learning stages in multiple regions of the whisker system using multi-site tetrode recordings and miniscope imaging in the same behavioral paradigm. We demonstrate that this paradigm can be used to study learning mechanisms on the behavioral and neuronal level, including insight-like learning, rule

[1]Institute for Physiology and Pathophysiology, Heidelberg University, Heidelberg, Germany. [2]Institute for Anatomy and Cell Biology, Heidelberg University, Heidelberg, Germany. [3]Present address: Institute for Experimental Epileptology and Cognition Research, University of Bonn, Bonn, Germany. [4]These authors contributed equally: Filippo Heimburg, Nadin Mari Saluti, Josephine Timm. ✉e-mail: groh@uni-heidelberg.de

reversal and extinction as well as neural encoding of diverse internal and external world parameters.

## Results

### Tactile discrimination setup and learning paradigm

We developed an ethological paradigm in which freely moving mice learn to discriminate different aperture widths with their whiskers in order to collect food rewards. The behavioral apparatus included a linear track equipped with lick ports (LP) for reward delivery (condensed milk) and loudspeakers for punishment (white noise, 120 dB, random duration between 1 and 3 s), both situated at opposite ends of the track (see Table 1 for parts list). To access the LPs, mice had to navigate through motorized apertures of varying widths (Fig. 1a). To ensure adult mice interacted with the aperture using their whiskers during the passage, we estimated the relevant range of aperture widths to be between 45 and 25 mm. Training sessions were done with ambient lights turned off and LEDs (overhead LEDs, backlights, beam brakes) operating in the infrared (IR) spectrum (850 nm), ensuring that mice did not use visual cues to discriminate apertures.

The session started when the mouse crossed the middle IR beam for the first time (Fig. 1a) during the approach to either of the two apertures. A trial started when one of the outer IR beams was crossed (Fig. 1a). Upon palpating the aperture with their whiskers, mice indicated their choice by either licking or not licking at the LP (Supplementry Video 1-3). The general rule logic for scoring the choices and triggering the outcomes was as follows (Fig. 1b, Table 2): licking in response to the go aperture was considered a "hit" and resulted in reward delivery, while not licking was considered a "miss" and resulted in no outcome. Licking in response to the no-go aperture was considered a "false alarm (FA)" and resulted in punishment, while not licking was considered a "correct rejection (CR)" with no outcome. In a later training stage, we introduced a "neutral" aperture which was equidistant to the go and no-go aperture (i.e., 35 mm) and which resulted in no outcome and no scoring.

Prior to training, mice underwent controlled food intake for two days with a target body weight of 85–90% of the initial body weight, which was maintained throughout the later training stages to increase motivation. During this period, mice underwent handling and LP training in their home cage. LP training was conducted with a dummy LP connected to a syringe via a flexible tube, and condensed milk was dispensed manually upon licking. The training consisted of multiple training stages with stage-specific criteria required to progress to the next stage (Fig. 1c, Table 2). During the habituation stage, in which the apertures remained in the go-state, mice learned to alternate between the two LPs and collect rewards. Habituation was completed after four sessions and at least fifty lick trials, after which mice progressed to learn the initial rule, wherein one aperture was linked to reward and a second aperture was linked to punishment. After mice learned the initial rule, they were trained for eight more sessions with the additional neutral aperture. The neutral stage was followed by the reversed rule stage, with reward and punishment contingencies reversed and the neutral aperture removed. In the final stage, the learned behavior was extinguished by randomizing both the apertures and the outcome. The mice's ability to differentiate between aperture widths was assessed using the discriminability index d-prime (d') (Fig. 1d, first column).

Operation of the setup and data acquisition was controlled by Syntalos[22]. Behavioral data was written into an event list which contained the aperture state and data from the three IR beams and the two LPs (Fig. 1a). Data from the event list was used to calculate the performance (Fig. 1d, first column). Recordings from the two high-speed cameras above each LP were triggered when the mouse crossed the outer beam sensors and ended upon the mouse licking or crossing the center IR beam (Supplementary Video 1, 2). These videos were used to extract whisker-aperture contact times and whisker angles using

DeepLabCut[23] (Fig. 1d, second column). An overview camera continuously recorded the entire track (Fig. 1a, Supplementary Video 3) and was used for tracking head-movements as well as mouse location and speed. Syntalos additionally operated and archived the multichannel electrode recordings and miniscope calcium imaging data (Fig. 1d, last two columns).

### Mice learn to discriminate apertures with their whiskers and learn different task rules

We trained a cohort of 12 mice on the basic training scheme as illustrated in Fig. 1c. First, we examined lick rates for the different apertures across all subjects over the course of learning. The analysis demonstrated a gradual increase in lick rates for the rewarded aperture and a steep decrease in lick rates for the punished aperture (Fig. 2a). This divergence in lick rates during the initial learning stage indicates the onset of the learning process, which is also reflected in the population success rate (Fig. 2b). Individual analysis revealed that 5 of the 12 animals increased their success rates in no-go trials by a factor of five at around 60% of the stage progression, suggesting a moment of insight previously described by Rosenberg et al.[24]. On average, mice needed 471 trials (±128) to surpass the expert-level performance threshold (d' = 1.65) (Fig. 2c). For extended analyses of d' and success rates over sessions see Fig. S1a, b. Throughout the learning process, mice more than tripled their activity rate from approximately 20 trials per session in the first four sessions to 60 trials per session in the last four sessions (Fig. S1c).

After reaching the expert-level criterion (4 sessions and 200 trials above expert threshold), mice were trained in the neutral stage, where the lick rates in response to neutral aperture trials were intermediate between the go and no-go lick rates (Fig. 2a, middle), suggesting that the mice can differentiate between more than two stimuli.

In the reversal stage, licking at the wide aperture resulted in punishment and licking at the narrow aperture resulted in reward. During the reversed rule stage, mice initially exhibited inverted lick rates, reflecting their adherence to the previously learned rule. Over time, they gradually adapted to the reversed rule, resulting in a shift toward the correct behavioral response. Notably, in both the initial- and the reversed rule stage, lick rates for the punished aperture dropped after approximately 60–70% of the learning progression (Fig. 2a), consistent with insight-like learning in the reversed stage (Fig. S1d). Notably, mice required more than twice the number of trials to learn the reversed rule in comparison to the initial rule (471 trials ± 127 vs. 1103 ± 256 trials, Fig. 2d). In agreement with slower reversal learning, the learning speed, estimated from the slope of a logistic fit of the d' progression, was more than two times slower compared to the initial learning (Fig. S1e).

We then investigated whether mice (n = 6) demonstrate faster relearning of previously learned rules. Indeed, during repeated rule reversals, mice relearned the rules within fewer trials compared to the first reversal (mean ± standard deviation, 1st reversal: 1078 ± 330 trials, 2nd reversal: 728 ± 144 trials, 3rd reversal: 701 ± 146 trials, Fig. 2d,S1f), indicating increased efficiency in relearning repeated rule reversals.

After reaching expert criterion in the reversed rule stage, mice underwent an extinction stage in order to extinguish the previously learned stimulus-outcome associations by randomizing both aperture states and outcomes in each trial (50% chance for "wide", "narrow", "reward" and "punishment"). During the extinction stage, mice changed their behavior within the first session, such that they always licked, regardless of the aperture state and outcome, resulting in a drop in the discrimination performance to chance level (Fig. 2e,S1g).

Further, we investigated the minimal aperture contrast that mice can robustly discriminate. We trained a cohort of eight animals on the previous stimulus set which has a contrast of 20 mm (wide = 45 mm; narrow = 25 mm). Subsequently, the contrast was reduced by steps of 4 mm and then by 2 mm. Animals

were required to reach the expert-level criterion at each contrast before progressing to the next smaller contrast. We determined the smallest discriminable contrast at 6 mm, below which

performance dropped below the expert-level performance threshold. At a contrast of 2 mm, performance dropped to chance level (Fig. 2f). Next, we trained three additional naive cohorts,

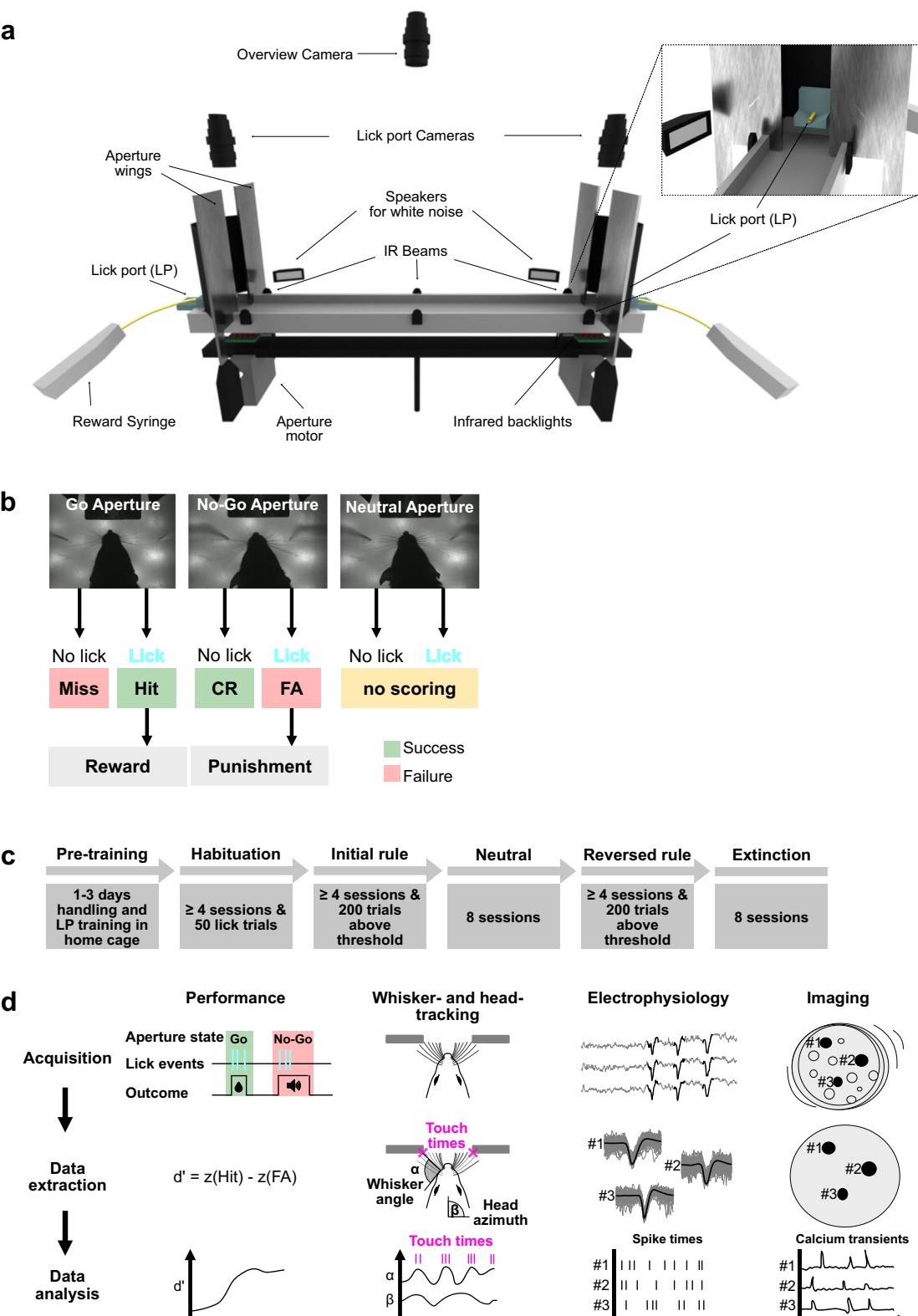

**Fig. 1 | Tactile discrimination setup and learning paradigm. a** Display of the whisker-discrimination setup, created with Blender v2.9 (files can be found here: https://github.com/GrohLab/A-tactile-discrimination-task-to-study-neuronal-dynamics-in-freely-moving-mice[41]). **b** Rule logic for scoring the choices and triggering the outcomes. **c** Training scheme with different learning stages and criteria to move to the next stage. **d** Overview of data acquisition and analysis pipeline for the analysis of task performance, whisker-aperture touch times, head-tracking, single-unit spike times and calcium transients.

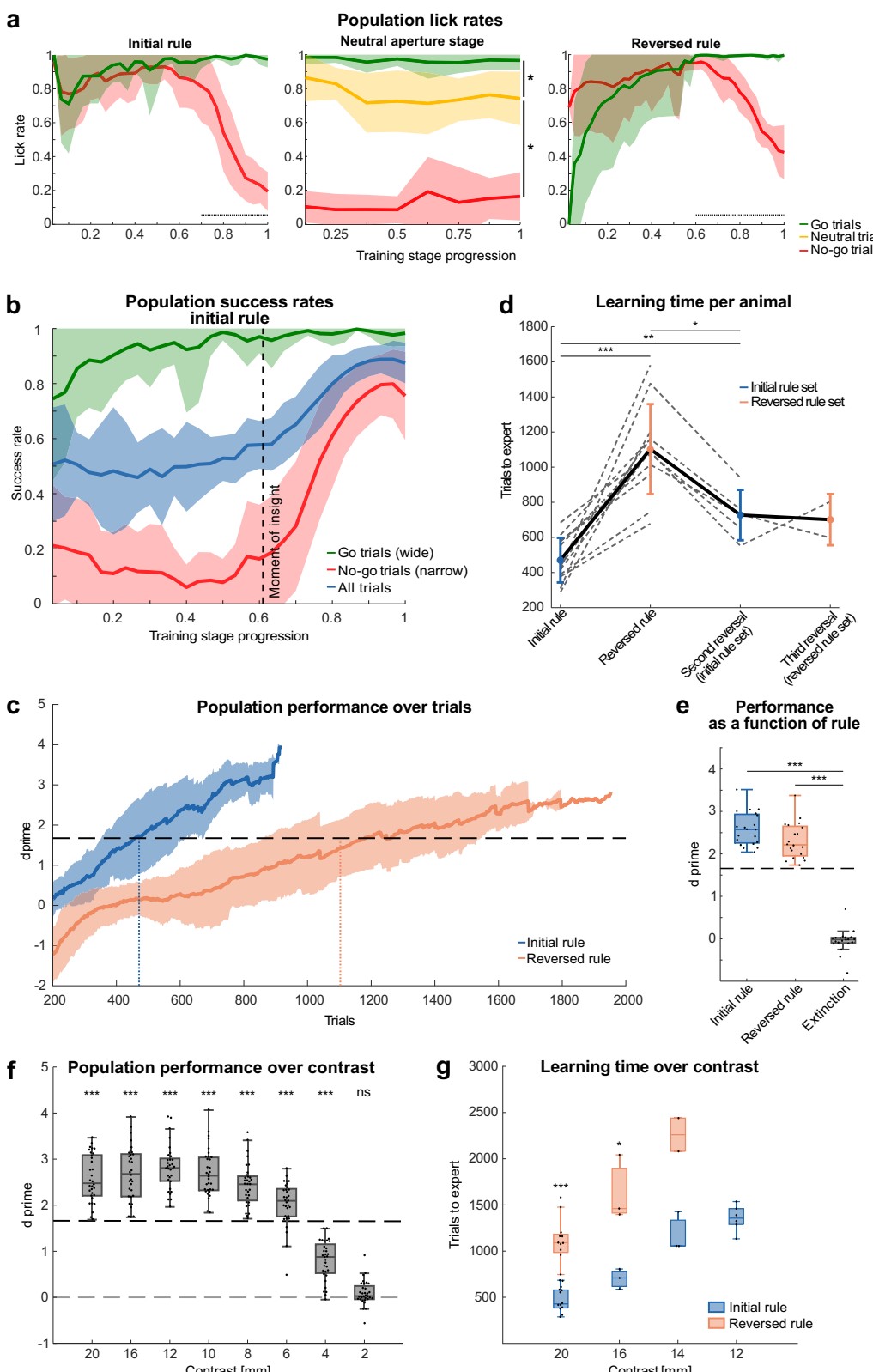

each on a different contrast (12, 14, 16 mm), in order to determine the learning times for different contrasts. We found that animals trained on a larger contrast needed fewer trials to learn the rules compared to animals trained on a smaller contrast (Fig. 2g,S2). For the 16 mm contrast, mice required an average of $701 \pm 110$ trials (Fig. S2a, blue line) to reach expert-level performance, whereas those trained on 14 mm and 12 mm contrast required

$1180 \pm 215$ trials (Fig. S2b, blue line) and $1356 \pm 142$ trials (Fig. S2c, blue line), respectively.

We next tested reversal learning for different contrasts. Again, compared with the initial rule stage, animals required more than twice as many trials to relearn the rule during the reversal stage (factor = 2.22, taking all contrasts into account). The factor between contrasts was not statistically different, suggesting that mice require about twice

**Fig. 2 | Mice learn to discriminate apertures with their whiskers and learn different task rules. a** Mean lick rates and standard deviation (bands) from $n = 12$ mice across three different rule stages. Horizontal dashed lines indicate significant differences. **b** Mean success rates and standard deviation (bands) over stage progression for Go- (Hit/(Hit+Miss)), No-go (CR/(CR + FA)) and all trials ((Hit+CR)/(Hit +Miss+CR + FA)) ($n = 12$ mice). Dashed line indicates "moment of insight", defined by a jump in success rate in no-go trials by a factor of five from one session to the next ($n = 5$ mice). **c** Mean performance and standard deviation over trials for 20 mm contrast. Solid lines show mean d', calculated with a running window of the preceding 200 trials; shaded areas show standard deviation, horizontal dashed line indicates expert-level performance threshold (d' = 1.65); $n = 12$ mice). Animals reached expert-level performance after 471 trials in the initial rule stage and after 1103 trials in the reversed rule stage (dotted vertical lines). **d** Learning times for initial-, 1st-, 2nd- and 3rd- rule reversal (dotted lines: individual animals; solid lines:

means and standard deviation; $n = 12$ mice). **e** Performance for initial rule, reversed rule and extinction. Horizontal dashed line indicates expert-level performance threshold (d' = 1.65). Performance was calculated from the last four sessions in the initial rule and reversed rule stage and the first four sessions in the extinction stage, each dot represents one session, $n = 5$ mice. **f** Population performance for mice trained on a gradually decreasing contrast. Horizontal dashed lines indicate chance-level and expert-level performance thresholds (d' = 0 vs 1.65, respectively). Performance was calculated from the last four sessions in each contrast condition, each dot represents one session, $n = 8$ mice. **g** Number of trials required to reach expert-levels for animals trained on different contrasts in the initial and reversed rule. ns: $p > 0.05$, *$p \leq 0.05$, **$p \leq 0.01$ and ***$p \leq 0.001$; **a** Welch's $t$-test, **d, g** paired $t$-test, **e** repeated measures anova, **f** one-sample $t$-test. For exact statistics and box plot definitions see Supplementary Table 1.

as many trials for reversal learning independently of the contrast. Specifically, during the reversal stage, mice required $1633 \pm 356$ trials to relearn the task with a 16 mm contrast (Fig. S2a, orange line) and $2260 \pm 255$ trials with a 14 mm contrast (Fig. S2b, orange line).

In summary, this paradigm can be used to train mice to discriminate apertures and to learn different task rules. Mice demonstrated the capacity to update their behavior to new reward-punishment contingencies during multiple reversal stages.

## Electrophysiological and optical recordings in freely moving mice during tactile discrimination

We explored the versatility of the setup to employ electrophysiological and imaging approaches in the same behavioral paradigm. Therefore, we first implanted tetrode arrays to record simultaneous unit-spiking activity from multiple areas of the somatosensory system in a cohort of 6 mice (Fig. 3a). Tetrode arrays were targeted to the barrel cortex (BC), ventral posteromedial nucleus (VPM), posterior medial complex (POm), and ventral zona incerta (ZIv) (Table 3). Analysis of the electrophysiological data in expert mice (d' ≥ 1.65), revealed significant modulation of neuronal firing rates following whisker-aperture contacts in all four brain regions (Fig. 3b). The onset latencies of these responses, measured as the time to the first significant deviation from baseline (see Methods for details), exhibited rapid recruitment of the recorded brain areas (Fig. 3c), supportive of correct targeting of the somatosensory pathway.

In addition to electrophysiology, we employed deep-brain calcium imaging (Fig. 3d). We chose the POm as the region of interest because of the recent attention on POm as a thalamic hub for higher cognitive functions[25–29]. To record in vivo calcium signals, we expressed GCaMP6f in POm (Fig. S3). Mice were then trained to reach expert-level performance and somatic calcium signals were imaged with miniscopes attached to an implanted GRIN lens (recording examples shown in Fig. 3e).

To compare neural activity across methods, we analyzed population peristimulus time histograms (PSTHs), which revealed differential responses to go and no-go trials (Fig. 3f). In line with that, we found several touch-modulated units that selectively responded to either one of the two aperture stimuli (Fig. 3g). Touch-modulated units comprised 63.2% in the BC (261 out of 413 units, with 142 responsive to the wide aperture, 28 to the narrow aperture, and 91 to both), 46.7% in the VPM (196 out of 420 units, with 90 responsive to the wide aperture, 53 to the narrow aperture, and 53 to both), 49.4% in the POm (129 out of 261 units, with 70 responsive to the wide aperture, 23 to the narrow aperture, and 36 to both), and 45.6% in the ZIv (67 out of 147 units, with 23 responsive to the wide aperture, 9 to the narrow aperture, and 35 to both). Compared to the electrophysiology results, we found a similar proportion of touch-modulated units in the calcium imaging data of POm (48.2%, 78 out of 162 units), with 34 responsive to the wide aperture, 31 to the narrow aperture, and 13 to both.

Furthermore, a convolutional neural network was trained to classify calcium transients over the whisker-aperture contacts (see Methods). The classifier could reliably distinguish the aperture width based on the neural calcium transients with the validation accuracy of 87.65% (Fig. S3b, c).

## Bilateral whisker input and barrel cortex are necessary for normal task performance and learning

Inspired by the popular texture-discrimination paradigm for rodents, our initial task design used sandpapers. However, we found that in our freely-moving configuration, mice exploited olfactory cues for sandpaper discrimination, evidenced by undisrupted performance when the whiskers were plucked or the sandpapers were reversed to the textureless backside. Subsequent NMDA lesions in the olfactory bulb completely abolished task performance (Fig. S4a), confirming that olfaction alone can drive discrimination in texture-based paradigms.

In contrast, the aperture-based task is purely whisker-dependent, as demonstrated by the following experiments. First, we performed bilateral whisker plucks in expert mice. The complete removal of whiskers deteriorated the performance of expert animals below performance threshold (Fig. 4a). In a separate experiment, animals with intact whiskers continued to perform at expert level with IR backlights and IR overhead LEDs turned off, demonstrating that visual cues were not used to solve the task (Fig. S4b). These results show that the employment of apertures with variable widths, eliminates the possibility for mice to use olfactory cues, ensuring that discrimination relies solely on whisker-mediated tactile processing.

To investigate the requirement of bilateral whisking we trained an additional cohort on 20 mm and 6 mm contrasts to expert-level performance, followed by unilateral whisker plucking. The performance dropped in all mice, however, only for the 6 mm contrast the performance dropped under the expert-level threshold (Fig. 4b). Similarly, when removing one of the aperture wings on each side of the track, mean population performance dropped below the expert-level performance threshold even for 20 mm contrast (Fig. S4c). Together these results demonstrate that bilateral whisking is required for smaller contrasts but animals might be able to use other facial parts to discriminate bigger contrasts.

We next addressed whether the BC is required for normal task execution and task learning by ablating the BC bilaterally similar as in ref.[19] (Fig. 4d Inset). Sham lesions served as control cohorts. In the first experiment, we addressed the BC-dependency for task execution with BC ablations in expert animals (Fig. S4d). Ablated animals, but not sham controls dropped below the performance threshold, demonstrating that task execution is BC-dependent (Fig. 4c). In a second experiment, we performed BC ablations and sham operations in naive animals and trained them on the basic task rule. While sham controls showed normal learning times (~500 trials to reach expert level), the learning in BC-ablated animals was severely impaired, with a roughly four-

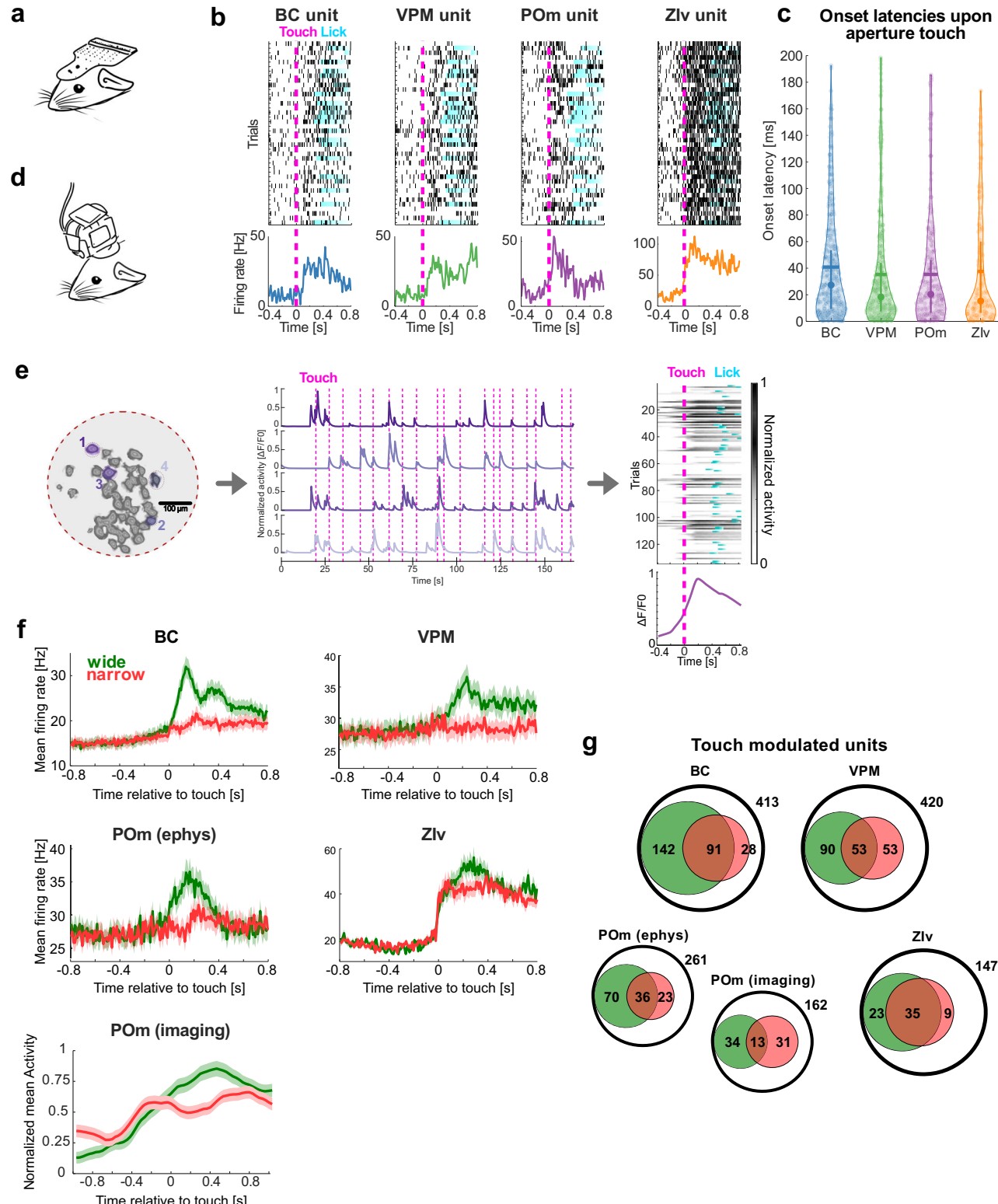

fold increase in trials required for expert level performance (Fig. 4d). Furthermore, the performance of BC-ablated animals remained marginally above expert-level performance and never reached similar d' levels as sham animals (Fig. 4e), as reflected by the larger proportion of trial-by-trial performance drops (Fig. S4e) and broader distribution of residuals from the sigmoidal fit (Fig. S4f). This learning deficit can be explained by the learning-dependent increase in aperture-selective units in BC under normal conditions (Fig. 4f), supporting the notion that the

formation of a robust, stimulus-selective representation in BC is a critical component for achieving stable and high-level task performance.

Given these findings, we next sought to determine whether temporarily disrupting whisker input in expert mice would similarly impact performance. Therefore, we applied a local anesthetic (4% Lidocaine cream) to both whisker pads of expert mice. For control a neutral ointment was used. Performance dropped below the expert-level-performance threshold with lidocaine application (Fig. 4g). To

**Fig. 3 | Neural activity patterns during aperture discrimination task.**
**a** Schematic representation of a head-mounted EIB. **b** Representative raster plots of single unit spiking activity in BC, VPM, POm, and ZIv, aligned to the time of whisker-aperture contact (magenta line). Below each raster plot, corresponding PSTHs display mean firing rates. **c** Onset latencies of neuronal responses following whisker contact with the aperture across different brain regions. Latencies were computed based on the first significant deviation from baseline activity (see Methods). Violin plots depict the distribution of response latencies, with mean values highlighted as bars, and median with IQR as filled circles with whiskers. **d** Schematic representation of a head-mounted UCLA Miniscope V4. **e** Spatial footprints of POm units from one example imaging session (left), with four touch-modulated units highlighted in color. Example normalized calcium transients (ΔF/F0) from the four touch-

modulated POm units (middle), with magenta lines indicating whisker touches with the aperture. Trial-aligned calcium transients of a representative POm unit and averaged across trials below. Lick onsets are marked in blue. **f** Population PSTHs displaying the mean firing rate (± SEM) of different brain areas, for wide (green) and narrow aperture trials (red), over all expert sessions. Neural activity is aligned to the time of whisker-aperture contact. For POm, normalized calcium activity (±SEM) from 39 POm neurons detected in a single session is shown. **g** Venn diagrams showing proportions of touch-modulated units/neurons per brain region (from a total sample indicated by black circles). Units/neurons responding selectively to the wide aperture are shown in green, those responding to the narrow aperture in red. The intersection represents units responsive to both apertures.

---

determine whether this impairment was due to altered whisker kinematics, we used a Generalized Additive Model[30] classifier to decode aperture width from whisker angular trajectories (Fig. 4h for expert sessions). Decoding accuracy remained high in both lidocaine and control conditions and significantly exceeded chance level (Fig. 4i), suggesting that whisker motion itself was not disrupted by local anesthesia. Passive whisker deflection upon encountering the apertures of variable width (Fig. S4g for control sessions), likely contributed to the high decoding accuracy. However, when decoding aperture width from spike trains recorded across different brain areas, we observed a marked reduction in classification accuracy during lidocaine sessions compared to controls (Fig. 4j), demonstrating that sensory information processing was impaired at the neural level.

Together, these results show that both learning and task execution of the freely moving discrimination paradigm are dependent on intact whiskers and the BC. Furthermore, the evolution of aperture-selectivity in BC during learning critically contributes to fast and stable learning.

### Capturing naturalistic decision-making, active touch strategies, and dynamic behavioral states during learning

In head-fixed go/no-go paradigms, licking is the primary decision variable. However, in our freely moving paradigm, an earlier decision-revealing movement—the mouse's approach or withdrawal from the aperture—may provide additional insights into decisions and learning. Indeed, during CR trials, mice exhibited a turning behavior, which we used to estimate reaction time (time between aperture touch and turning) and reaction distance (how close the mouse approached the LP before turning). Both measures closely correlated with performance: expert mice showed significantly shorter reaction times and distances, indicating a refined and more efficient decision-making process (Fig. 5a–c).

Moreover, we observed a refinement of licking behavior. While naive mice exhibited comparably late and more variable lick latencies following whisker-aperture interaction, expert mice showed earlier and more reliable licks (Fig. 5d for a representative session). Population analyses confirmed this trend, showing a systematic decrease in lick latencies and variability with learning (Fig. 5e).

To assess active touch strategies, we analyzed whisker kinematics and their relationship to aperture contact events. Whisker tracking revealed a rhythmic pattern of whisking at $14.5 \pm 3.1$ Hz (mean ± std) (Fig. 5f), consistent with previously described whisking frequencies in freely moving mice[31]. Interestingly, while the number of whisking cycles remained stable in FA trials, they significantly decreased in CR trials as mice became experts (Fig. 5g).

Heat maps of positional data illustrate that mice preferentially occupied the sampling and reward areas, while avoiding exposed regions (Fig. 5h). This pattern may reflect a strategic balance between exploration and risk avoidance. Analysis of spatial tuning revealed a subset of putative spatially tuned units that exhibited distinct firing rate modulations in relation to position on the track (Fig. 5i), with significant proportions of such units found across multiple brain

regions ($2.4 \pm 2.1\%$ in BC, $6.2 \pm 4.3\%$ in VPM, $8.7 \pm 3.9\%$ in POm, $11.5 \pm 12.7\%$ in ZIv; Fig. 5j).

We also found neuronal activity to be modulated by locomotor state. Comparing firing rates during rest and locomotion revealed a subset of units with locomotion-enhanced or locomotion-suppressed activity (Fig. 5k). Notably, significant increases in mean firing rates during movement were observed in VPM, POm, and ZIv on a population level.

In summary, the tactile discrimination paradigm can be utilized to study different motor behaviors over the course of tactile discrimination learning. The results demonstrate that learning manifests itself not only in increased tactile discrimination performance but also in adaptation of motor behaviors.

### Neural encoding of whisker- and head kinematics during active sensory exploration

To investigate the relationship between whisker kinematics and neural representations during task execution, we analyzed the modulation of firing rates by whisking parameters, including whisker angle and whisking phase. We observed a significant increase in mean firing rates during whisking compared with quiescence across all recorded brain regions (Fig. S5), emphasizing the neural engagement during active sensory exploration. We further identified a subset of roughly 10% of units ($10.5 \pm 2.2\%$ in BC, $9.8 \pm 10.6\%$ in VPM, $10.8 \pm 8.4\%$ in POm, $12.4 \pm 6.3\%$ in ZIv) across all four recorded brain regions—BC, VPM, POm, and ZIv—that exhibited significant modulation by whisking angle (Fig. 6a, b), as determined by comparisons to null distributions (see Methods). The proportions of these whisker angle-tuned units were comparable between VPM and POm, consistent with previous reports in both head-fixed awake[26] and freely moving mice[32]. Further examination of whisking phase tuning showed distinct subsets of units in each area ($33.2 \pm 11.5\%$ in BC, $39.3 \pm 15.0\%$ in VPM, $34.0 \pm 10.0\%$ in POm, $31.1 \pm 16.3\%$ in ZIv) that were modulated by the phase of the whisking cycle (Fig. 6c, d). Analysis of population polar plots revealed a preference for protraction states among phase-tuned units ($143.5°$ in BC, $176.8°$ in VPM, $179.8°$ in POm, $157.6°$ in ZIv; Fig. 6e), suggesting that neural representations of whisking dynamics are temporally structured and phase-dependent. Beyond whisker-based encoding, we examined whether neural populations also track head kinematics. We found that BC, VPM, POm, and ZIv carry information about head azimuth (Fig. 6f), in agreement with the recent observation of head-movement coding in POm and VPM[32]. Notably, while the overall proportion of head-angle tuned units was comparable across regions (Supplementary Table 1), ZIv exhibited a lower proportion of allocentric head-tuned units, though this difference was not statistically significant (p (ZIv vs. BC) = 0.0625; p (ZIv vs. VPM) = 0.4375; p (ZIv vs. POm) = 0.3125, two-sided Wilcoxon signed rank test, Fig. 6f).

## Discussion

This study introduces a go/no-go, whisker-dependent discrimination paradigm for freely moving mice. Unlike head-fixed approaches, our freely moving design enables investigations of mouse behavior and

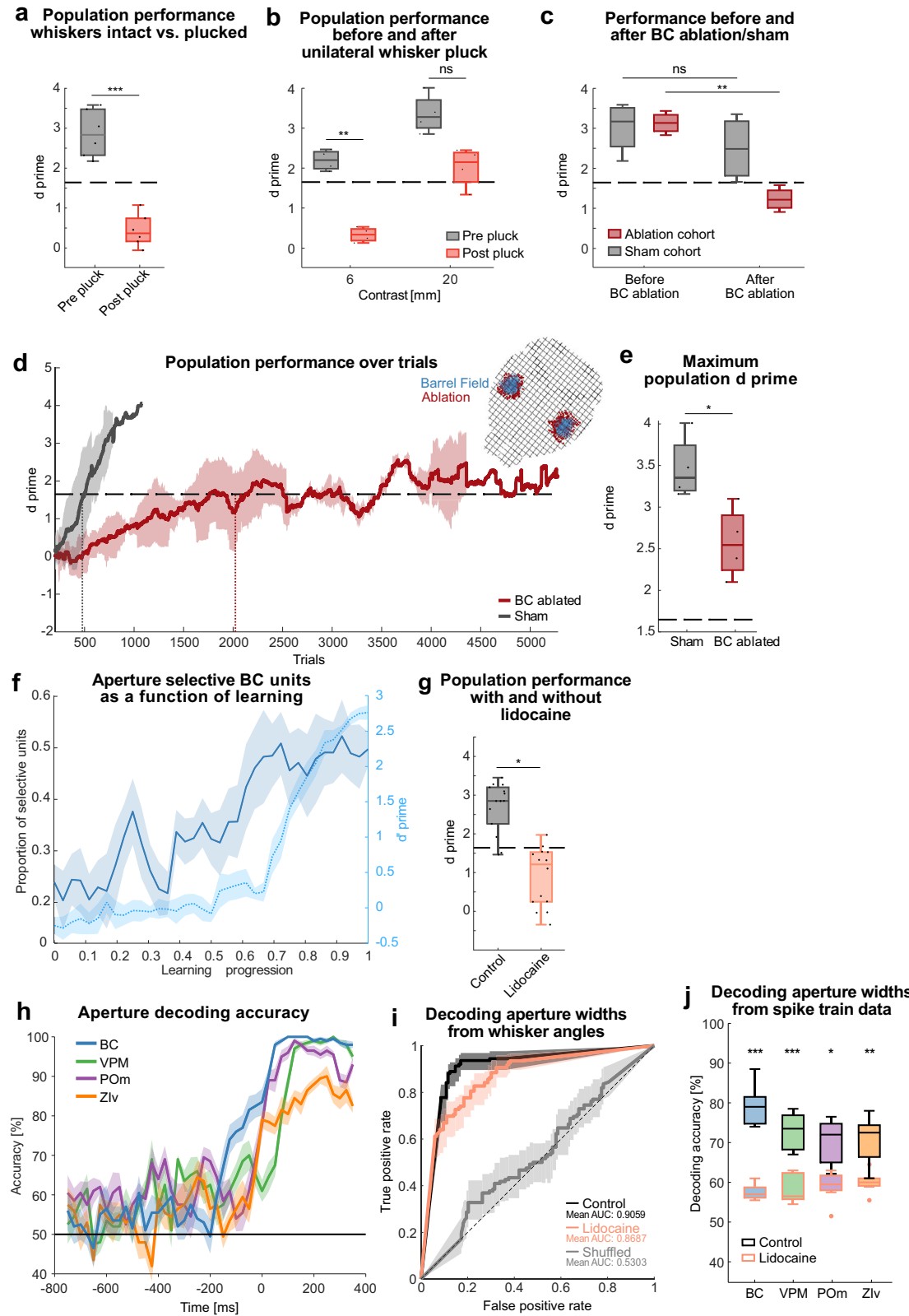

**a** Population performance whiskers intact vs. plucked

**b** Population performance before and after unilateral whisker pluck

**c** Performance before and after BC ablation/sham

**d** Population performance over trials

**e** Maximum population d prime

**f** Aperture selective BC units as a function of learning

**g** Population performance with and without lidocaine

**h** Aperture decoding accuracy

**i** Decoding aperture widths from whisker angles

**j** Decoding aperture widths from spike train data

learning in a naturalistic setting while remaining compatible with high-resolution electrophysiology, calcium imaging, and motor behavior tracking.

A key advantage of the unrestrained configuration is the ability to capture richer behavioral readouts. For instance, head movements–absent in restrained paradigms–can serve as decision variables alongside conventional lick metrics. Additionally, the clear temporal

separation between sensory sampling and reward consumption reduces confounding lick-related activity during perception. In contrast, anticipatory licking in head-fixed paradigms often overlaps with sensory processing, complicating interpretation.

The discrimination task relies solely on somatosensory whisker information for aperture discrimination, demonstrated by deteriorated performance when whiskers were anesthetized or

**Fig. 4 | Intact whisker input and barrel cortex are necessary for normal task performance and learning. a** Population performance for expert animals before and after bilateral whisker-pluck. Performance calculated from last session before and first session after the whisker pluck, each dot represents one session, dashed line indicates expert-level threshold (d′ = 1.65), $n = 6$ mice. **b** Same as in a but with unilateral whisker pluck and for contrasts of 6 and 20 mm ($n = 4$ mice for each contrast). **c** Same as in a but with barrel cortex ablations (red) or sham operations (gray). **d** Performance over trials in barrel cortex ablated mice (red) and sham operated animals (gray). Expert-level performance (d′ = 1.65, gray horizontal dashed line) was crossed after 478 vs. 2023 trials for sham (gray vertical dashed line) and ablated mice (red vertical dashed line), each $n = 4$ mice. Inset: example reconstruction of barrel cortex ablation. **e** Population performance for expert animals in d, calculated from the session with the maximum d prime for each mouse ($n = 4$ mice each). **f** Proportion of aperture-selective BC units across learning progression (solid line, mean ± SEM). Aperture selectivity was assessed using a two-sided Wilcoxon signed-rank test, comparing unit firing rates between the two apertures. The dotted light blue line represents the average d-prime (± SEM), $n = 6$ mice. **g** Population performance for expert animals ($n = 4$ mice) with alternating application of lidocaine or neutral ointment (control) to both whisker pads.

Performance was calculated from 7 lidocaine and 7 control sessions across all mice for each condition. Same conventions as in a. **h** Time-resolved decoding accuracy and standard error of mean (bands) of aperture width from spike data of different brain areas. Spike data was analyzed from expert sessions (800 ms before to 400 ms after aperture touch, $n = 6$ mice). **i** Receiver operating characteristic curves from a generalized additive model classifier based on whisker angles after whisker-touch onset. Comparison of control (black), lidocaine sessions (orange), and randomly shuffled labels (light gray). Accuracies for control and lidocaine sessions were not significantly different from each other, but both differed significantly from the shuffled accuracies ($n = 3$ mice). **j** Decoding accuracy from spike data for lidocaine sessions (orange frames) and control sessions (black frames). The mean decoding accuracy was calculated over a time window from trigger onset to 400 ms post-trigger onset (see Methods). Number of units (from $n = 3$ mice) in lidocaine sessions: 128 (BC), 100 (VPM), 52 (POm), 17 (ZIv); and control sessions: 150 (BC), 101 (VPM), 64 (POm), 17 (ZIv). ns: $p > 0.05$, *$p ≤ 0.05$, **$p ≤ 0.01$ and ***$p ≤ 0.001$; **a, b** two-tailed paired $t$-test, **c** two-way anova, **e** independent samples $t$-test, **g** repeated measures anova, **j** two-tailed Wilcoxon rank sum test. For exact statistics and box plot definitions see Supplementary Table 1.

removed. Furthermore, compared to texture discrimination tasks[6,17,33–36], in which surface differences are created by switching between different sandpapers, the use of the same apertures with variable widths ensures the elimination of potential confounders from olfactory cues.

Mice learned the task rapidly and reliably, typically reaching expert-level performance within ~4 h of training across eight days. This fast learning not only allows for high-throughput behavioral analysis but also provides several other advantages. First, the paradigm allows the incorporation of multiple consecutive learning stages in which mice learn different rules (initial, neutral, reversal, devaluation, within a total of 29 ± 4 days). Second, the paradigm can be used in repeated rule switch mode, by switching back and forth between two inverted rule sets (Fig. 2d). This can be advantageous for learning studies that require a steady-state learning environment without confounding learning factors which happen only in the initial learning when mice are exposed to the learning paradigm for the first time. Indeed, we found several behavioral adaptations, which happen in the initial learning and then stabilize, for example, overall activity (trial counts per session), lick latencies, and overall learning speed, which progressively approached an efficient optimum following multiple rule switches. Most importantly, the fast learning allows for chronic neurophysiology within a short experimental time frame. This is advantageous to monitor neuronal activity over the entire learning period (Fig. 4f) and even during multiple learning rules, which is often difficult to achieve because of the deteriorating signal quality and unit yield in chronic electrophysiological and optical recordings[37–39]. For instance, in a parallel study, we successfully utilized the learning platform to monitor neuronal dynamics over multiple learning and reversal stages to investigate neuronal encoding of reward contingencies during learning[40].

The setup is robust, modular, and largely automated. Once configured, experiments can run without the experimenter in the room, reducing potential biases. Maintenance is minimal aside from routine cleaning after the experiments. All trained mice in our hands learned the task successfully, and detailed documentation[41] makes the system readily adoptable. The modularity of the hardware and acquisition software[22] allow easy customization to other research approaches, including integration with silicon probes, Neuropixels, optogenetics, or custom imaging setups.

This versatility enables the platform to address a wide range of research questions. For example, it can be used to study spatial navigation (Fig. 5i, j) and head-orientation (Fig. 6f), or be adapted into a delay-based task to probe short-term memory mechanisms[42,43] by increasing the distance between the aperture

and the lick spout. The paradigm also allows investigation of perceptual uncertainty by training mice at near-threshold contrasts (Fig. 2f), extending work on sensory discrimination under ambiguous conditions[44–47]. Furthermore, it is suitable for probing how internal states such as sleep, hunger[48], or social experience influence learning, as well as for examining the contribution of specific neural circuits—such as the higher-order thalamic nucleus POm (Fig. 3)—to task performance[27,28,40].

During early learning and extinction phases, mice often displayed constitutive licking across all trials, regardless of aperture width. This behavior, which produced a 50% reward/punishment outcome, indicates that early performance was driven by reward motivation before stimulus-outcome associations were established. As learning progressed, mice flexibly updated their behavior in response to rule changes, demonstrating cognitive flexibility. Notably, previously learned rules were re-acquired more quickly (Fig. S1g), consistent with the phenomenon of "savings" observed in human studies[49,50].

Mice reliably discriminated aperture widths down to approximately 6 mm—a threshold similar to that reported in rats using forced-choice paradigms[20]. However, given sufficient motivation, mice likely achieve even finer discrimination, as they have been shown to detect submillimeter differences in sandpaper textures[34]. In contrast to texture discrimination tasks, where the whiskers interact with fine surface features via stick-slip micromotions[36,51–53] our distance discrimination task likely depends more on whisker-angle changes.

Our data also show that both bilateral whisker input and intact barrel cortex are necessary for normal task performance and learning speed. These findings align with previous studies demonstrating cortical dependence in whisker discrimination tasks[35]. To investigate how the barrel cortex supports learning, we performed longitudinal recordings and found that aperture-selective neurons emerged over time in the barrel cortex (Fig. 4f). This suggests that the development of robust, stimulus-specific representations in the cortex is a key factor enabling efficient learning and stable performance.

Electrophysiological and optical recordings revealed touch-modulated responses across multiple brain regions, including BC, VPM, POm, and ZIv. Touch modulation has been reported in studies investigating these regions mostly separately and under head-fixation[8,54–57]. Here, by recording all four regions simultaneously in freely moving animals, we report sequential touch representation, with surprisingly early touch responses in the ZIv (Fig. 3c), consistent with the idea of feed-forward inhibition of POm via the ZIv[58,59].

Aperture width could be decoded from spike trains or calcium signals, and decoding was significantly impaired by lidocaine

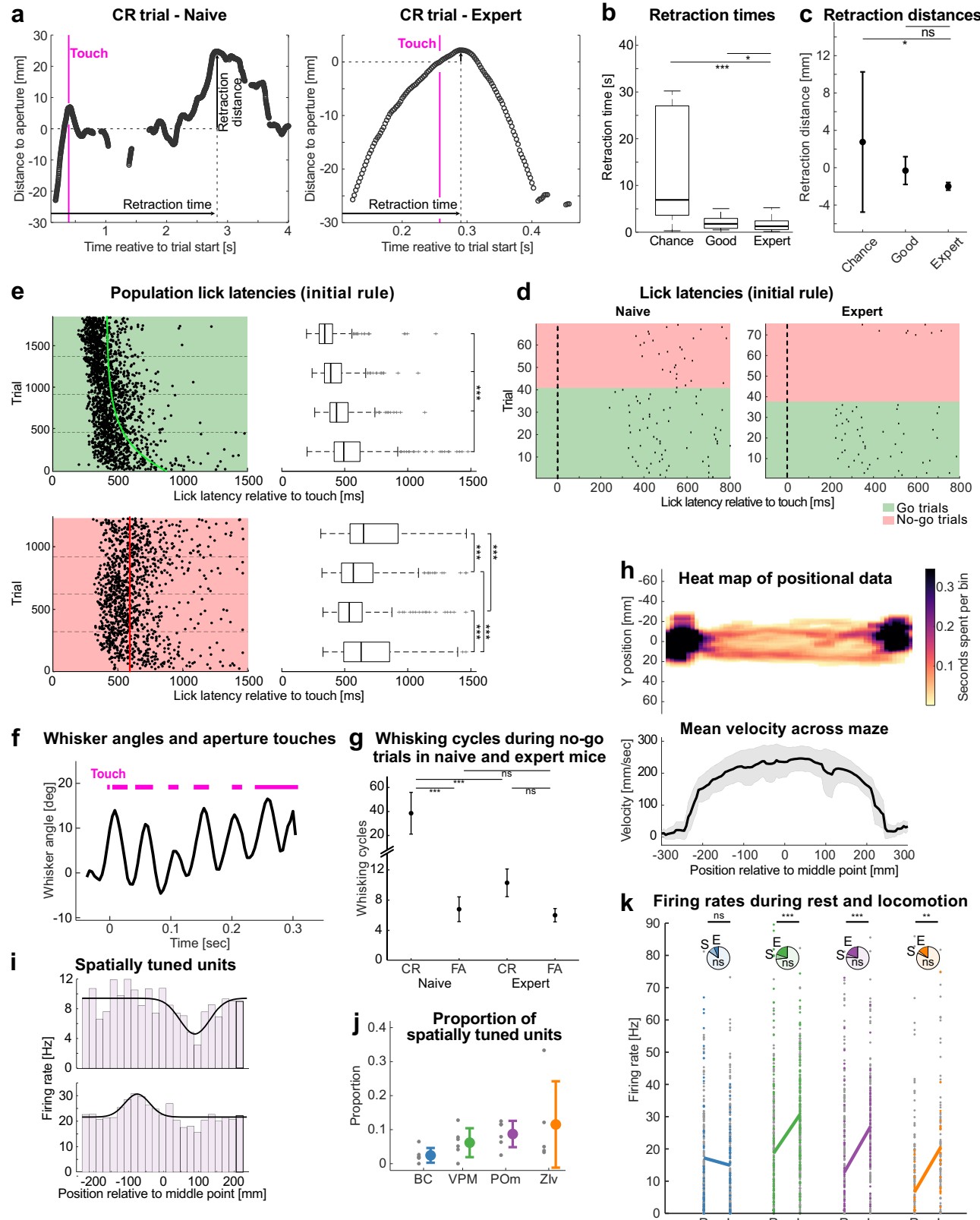

anesthesia of the whisker pad. These results further support that task-relevant sensory information is encoded in the thalamocortical system.

Importantly, neuronal activity was modulated by behavioral state and movement-related variables, including locomotion (Fig. 5k),

whisking (Fig. S5), whisker angle (Fig. 6a, b), whisker phase (Fig. 6c–e), head angle (Fig. 6f), and spatial position (Fig. 5i, j). By characterizing this tuning in an unrestrained paradigm, our work provides a foundation to study how sensorimotor integration shapes perceptual decision-making in naturalistic conditions.

**Fig. 5 | Capturing naturalistic decision-making, active touch strategies, and dynamic behavioral states during learning. a** Approaching behavior to the lick spout following whisker contact ("Touch") with the aperture in a CR trial, shown for a naive session (left) and an expert session (right). The blue trace represents the snout trajectory over time. Retraction time and distance are quantified from the maximum of the trajectory (= turning point), with respect to the beam break (retraction time) and aperture position (retraction distance). **b** Retraction times across different learning stages (chance, good, expert). **c** Same as b for Retraction distances (mean with 95%CI). **d** Example lick latencies from one mouse after whisker-aperture interaction (t = 0 ms) for an early (naive, left) and an expert session (right). Go trials are highlighted in green, no-go trials in red. **e** Population lick latencies (n = 6 mice) following whisker-aperture interaction (t = 0 ms) across the initial learning stage for Hit (green) and FA (red) trials. Trials are grouped into four equal bins based on trial progression for statistical comparison between early (bottom) and late (top) trials. Trendlines (green and red) represent the best-fit asymptotic function (see methods). **f** Example whisker angle and aperture touches (magenta) during a hit trial (averaged angle values α from four DeepLabCut-tracked whiskers). **g** Number of whisking cycles during CR and FA trials (mean with 95%CI)

in naive (initial 4 sessions) and expert sessions (final 4 sessions). **h** Heat map of positional data along the linear track of the setup of an exemplary session. The color indicates the time spent in that area (see Methods). The corresponding mean velocity and standard deviation (band) across the track is shown below. **i** Representative examples of spatially tuned units, showing firing rates as a function of position relative to the middle point (mm), with firing rate histograms and fitted Gaussian tuning curves (black). **j** Proportion of spatially tuned units across all recorded brain regions. Each dot represents the mean proportion for an individual animal, with the cohort mean ± standard deviation as colored bars. **k** Firing rates during rest (R) and locomotion (L), across brain regions. Each dot represents the firing rate of a single unit (non-significant units in gray, significant units in color). Pie charts indicate the proportion of significantly modulated units (E: enhanced activity; S: suppressed activity). Solid lines connect mean firing rates during R and L for significantly modulated units. ns: $p > 0.05$, $*p \leq 0.05$, $**p \leq 0.01$ and $***p \leq 0.001$; **b**, **c**, **g** two-tailed Wilcoxon rank sum test, **e** Kruskal–Wallis Test, **k** two-tailed paired $t$-test; $n = 413$ (BC), 420 (VPM), 261 (POm), and 147 (ZIv) units in total. Data was pooled from $n = 6$ mice. For exact statistics and box plot definitions see Supplementary Table 1.

## Methods

### Ethics Statement
All experimental procedures were approved by the local governing body (Regierungspräsidium Karlsruhe, Germany, approval numbers: 35-9185.81/G-216/19 and 35-9185.82/A-8/20) and performed according to their ethical guidelines.

### Animals
The experiments were done with adult male mice from the inbred strain C57BL/6NRj (Janvier Labs, Le Genest-Saint-Isle, France). Mice were 8–10 weeks old at the start of training, with a mean body weight of 25.94 ± 2.33 g (mean ± standard deviation) at the start of the experiments. Mice were separately housed in a ventilated Scantainer (Scantainer Classic, SCANBUR A/S, Karlslunde, Denmark) under a 12-h inverted light/dark cycle (lights off at 7:00 a.m. and on at 7:00 p.m.) at a controlled temperature (20–24 °C) and humidity (45–65%). During behavioral training, mice were held on a controlled food intake schedule, and their body weight was maintained at 95–85% of their initial body weight. Under the controlled food intake schedule, mice were given dry food pellets (10–20% of their body weight) daily at the end of behavioral training. Water was available ad libitum.

### Behavioral apparatus
Behavioral testing and recording were conducted on an elevated, dark-toned linear platform (Table 1) measuring 82 cm in length and 10 cm in width (technical drawings are available on https://github.com/GrohLab/A-tactile-discrimination-task-to-study-neuronal-dynamics-in-freely-moving-mice/tree/main/MaterialList)[41], featuring a lick port (LP) at both ends. To access the LPs, mice were required to traverse through an aperture between two aluminum wings mounted onto a motor to adjust the distance between the wings (= aperture width) in each trial. Each LP consisted of a cannula attached to a syringe filled with condensed milk. A piezo sensor, mounted on top of the cannula, detected vibrations and registered lick events. A speaker was mounted close to the LP to deliver white noise. A lick event resulted in either reward delivery (10 μl of condensed milk) or punishment (120 dB white noise, random duration between 1 and 3 s) (Fig. 1b). Moreover, three pairs of infrared (IR) sensors were placed along the track, one in the center of the track and the other two in front of the apertures. Three cameras were positioned above the setup: an overview camera (60 Hz frame rate), providing a view of the entire platform, and two high-speed cameras (240 Hz frame rate) capturing the aperture and LPs for later analysis of whisker movements and touch events (Fig. 1a, Supplementary Videos 1–3). The behavioral apparatus was controlled using Syntalos (Code available at https://github.com/syntalos/syntalos), which controlled the devices and recorded the timing of

events (i.e., aperture state, reward/punishment administration, lick detection, and beam crossings of the mice), cameras, electrophysiology, and miniscope signals.

### Behavioral paradigm
Mice were trained to perform a go/no-go whisker-dependent paradigm on a linear track (Fig. 1), in which they had to learn to discriminate between apertures of different widths to get a reward and avoid punishment. Mice had to choose the correct response depending on the presented stimulus. The response can either be a lick response (hit for go trials, false alarm [FA] for no-go trials respectively) or to refrain from licking (crossing of middle beam brake) and move on to the subsequent trial (miss for go trials, correct rejection [CR] for no-go trials respectively). Animals were trained on the setup twice per day, each training session lasting 15 min, with a break of at least 2 h between the two sessions. At the beginning of the session, the mouse was put on the setup at the middle IR beam (Fig. 1a) to start the session. The mouse then runs to one end of the linear track. A trial started when one of the outer IR beams was crossed (Fig. 1a) during the approach to the aperture. Upon sampling the aperture with its whiskers, the mouse has to decide to either lick at the LP or turn and go to the other side of the linear track. A trial ends either after a lick response or, in case the mouse didn't lick, when the mouse crosses the middle point on its way to the other LP. At the beginning of each trial (crossing one of the outer IR beams), the aperture state on the opposite side end of the track was reset randomly. The apertures also perform mock movements when the same aperture was presented twice in a row to avoid any learning due to auditory cues. Table 2 summarizes the stimulus-outcome rules for the different training stages.

### Training procedure
The training procedure consisted of several stages, and mice must reach a predetermined criterion in each stage to move on to the next stage (Table 2). <u>Controlled food intake, handling, and LP training:</u> One week prior to behavioral training, the mice went onto a controlled food intake schedule while being handled and trained with the LP. During LP training, a dummy LP was placed in the mouse's home cage, which mimicked the appearance and functionality of the actual LP used in experimental tasks. Whenever the mouse approached the LP to lick, the experimenter manually dispensed condensed milk, as positive reinforcement to associate licking with receiving a reward. <u>Habituation stage:</u> Milk rewards were presented upon licking alternatingly at each LP in order to encourage mice to run back and forth between the two LPs. Apertures remain in the wide position. Once the mice reached criterion (Table 2), they progressed to the initial rule stage. <u>Initial rule stage:</u> Go (45 mm "wide") and no-go (25 mm "narrow") apertures were

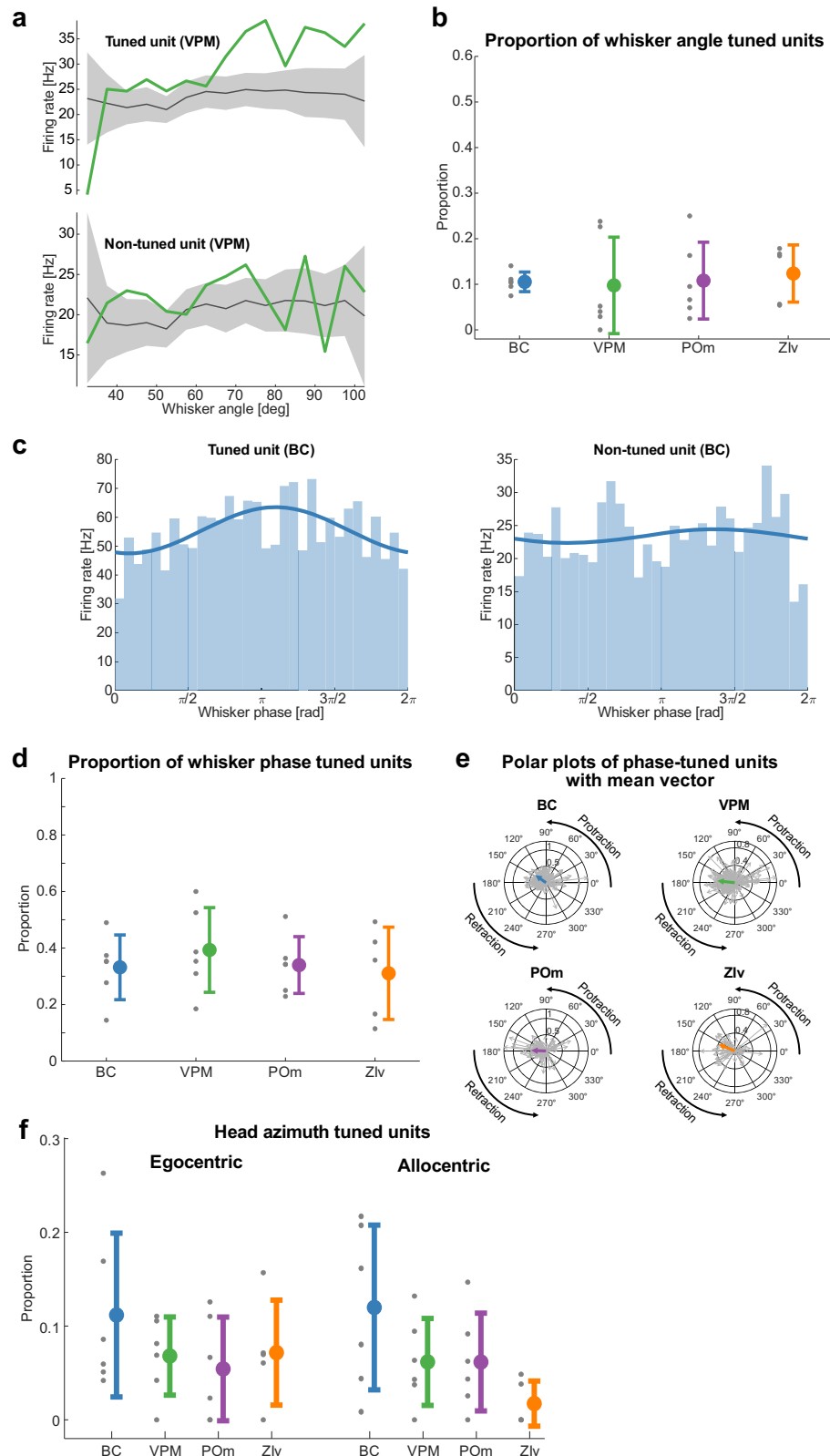

presented with an equal probability of 50% in a random sequence. Licking upon the go (wide) aperture triggered a reward, and licking at the no-go aperture triggered a punishment (Table 2). Individual mice completed the initial rule stage at an individual point in time, defined by the expert-level criteria (Table 2). For the experiments that involved a gradual decrease in aperture contrast (Fig. 3a), a new aperture width was introduced once the mouse reached the expert-level criterion or

when the mouse performed ≥ 4 sessions and ≥ 500 trials at a performance level below expert-performance threshold (d' = 1.65). The contrast was reduced to a minimum of 2 mm. Neutral aperture stage: Reward and punishment contingencies remained as in the initial rule stage, but an additional "neutral" aperture (35 mm) was added. Licking upon the neutral aperture resulted in no outcome and no scoring. The neutral aperture stage lasted for 8 sessions (Table 2). Reversed rule

**Fig. 6 | Neural encoding of whisker- and head kinematics during active sensory exploration. a** Representative examples of whisker angle tuning in the VPM, showing firing rates of a tuned unit (top) and non-tuned unit (bottom) as a function of whisker angle with null distributions (gray, mean ± SEM). **b** Proportion of units significantly tuned to whisker angle in the BC, VPM, POm, and ZIv (see Methods). Each dot represents the mean proportion for an individual animal, with the cohort mean ± standard deviation as colored bars. **c** Examples of whisker phase tuning in the BC, with firing rates plotted for a tuned unit (left) and a non-tuned unit (right) across whisker phases. A sinusoidal function is fitted to the respective distribution.

**d** Proportion of units significantly and standard deviation (bars) tuned to whisker phase in the BC, VPM, POm, and ZIv. Representations follow the logic of Fig. 6b. **e** Polar plots of phase-tuned units across brain regions, displaying individual vectors (gray arrows) and the mean population vector (colored arrows) of whisker phase tuning. **f** Proportion of units and standard deviation (bars) tuned to head azimuth with egocentric (left) and allocentric (right) references across BC, VPM, POm, and ZIv. Representations follow the logic of Fig. 6b. **a** Test against null-distribution of shuffled spike times (see Methods), **c** Kuiper test; $n = 413$ (BC), 420 (VPM), 261 (POm), and 147 (ZIv) units in total. Data pooled from $n = 6$ mice.

stage: Reward and punishment contingencies were inverted (i.e., the previously rewarded aperture was punished, and vice versa). The neutral aperture was not presented. Individual mice completed the reversed rule stage at an individual point in time, defined by the same expert-level criteria as in the initial rule stage (Table 2). Extinction stage: Reward and punishment were randomly presented, irrespective of the aperture state. The extinction stage lasted for 8 sessions (Table 2).

### Data acquisition

Data acquisition during behavior was done with Syntalos[22]. An intan-module recorded all analog signals from the LPs. The overview camera was connected to a recorder module that operated continuously throughout the session. The high-speed cameras and an event list were controlled by a custom-made Python script. The cameras were programmed to start recording when the animal approached the corresponding LP, determined by the crossing of the corresponding light beam. The event list kept track of the triggered light beams and reported the state of the apertures. The list also noted if a trial was a success or failure.

### Electrophysiological recordings

**Recording array.** Electronic Interface Boards (EIBs) were based on the design developed in ref.[60]. Tetrodes were manufactured in house by folding and twisting approximately 30 cm of tungsten wires (12.5 μm, Item # 100211, Tungsten 99.95% CS Wire, California Fine Wire Company). A total of 16 tetrodes (yielding 64 channels in total) were inserted to a custom-made EIB (produced by Multi-Circuit-Boards). The layout allowed for proper alignment of the tetrodes in the x-y plane. Each tetrode's placement was determined using a scaffold, ensuring precise targeting of BC (5 tetrodes), the VPM (4 tetrodes), the POm (4 tetrodes), and the ZIv (3 tetrodes). Coordinates were defined with respect to bregma, according to ref.[61] (Table 3). After insertion of the tetrodes into the EIB, tetrodes were fixed using UV-curable adhesive (3 M™ Filtek™ Supreme XTE, 3 M Deutschland GmbH, Neuss, Germany) and the tungsten wires were then soldered onto the copper traces of the EIB. The tetrodes were then suspended in a gold chloride solution (200 mg/dL gold chloride in deionized water, HT1004-100ML, Sigma-Aldrich, Inc., MO, USA) and gold plated to reduce their impedances to less than 100kOhm. To ensure proper conductance of the individual channels, a function generator (A365 Stimulus Isolator, World Precision Instruments, Inc., FL, USA) was connected to a basin filled with sodium chloride (120 mM), in which the tetrodes were submerged. Only EIBs with ≥62 conducting channels were used for implantation.

**EIB implantation.** 30 min before inducing general anesthesia, buprenorphine hydrochloride (Bupresol vet. Multidose 0,3 mg/ml, 10 ml, CP-Pharma Handelsgesellschaft mbH, Burgdorf, Germany) was administered subcutaneously at a dosage of 0.1 mg/kg of body weight. General anesthesia was induced using isoflurane (1–2% in oxygen, Isofluran Baxter, Baxter Deutschland GmbH, Germany), with control of the eyelid reflex and pedal withdrawal reflex. Body temperature was monitored and maintained between 36–38 °C. Eye ointment (Bepanthen®, Bayer, Germany) was applied to the eyes and saline solution

(30 ml/kg of body weight, Fresenius Kabi Deutschland GmbH, Bad Homburg, Germany) was administered subcutaneously. Lidocaine (Xylocaine® 1%, Aspen Pharma Trading Limited, Ireland) was injected under the scalp to provide local anesthesia in the surgical area. The animal was positioned in a non-traumatic stereotactic alignment apparatus (David Kopf Instruments, Tujunga, CA, USA). Small holes were drilled in the skull over the target areas (Table 3, coordinates relative to bregma) using a dental drill (78001 Microdrill, RWD Life Science, TX, USA). The tetrodes were then gradually lowered into the brain until the desired depth was reached. The EIB was secured to the skull using a cyanoacrylate-based adhesive (Super-bond, Sun Medical, Japan) and dental cement (Paladur®, Kulzer, Germany). Following the surgeries, animals were placed into their home cage on a heated surface for a minimum of 2 h before being returned to the ventilated Scantainer. Behavioral experiments started after an adequate period of recovery following the EIB implantation.

**Histology and identification of tetrode positions.** After administering a lethal dose of ketamine-xylazine through intraperitoneal injection (ketamine: 120 mg/kg bw, CP-Pharma Handelsgesellschaft mbH, Burgdorf, Germany; xylazine: 20 mg/kg bw, Xylavet® 20 mg/ml, CP-Pharma Handelsgesellschaft mbH, Burgdorf, Germany), a transcardial perfusion with paraformaldehyde (PFA, 4% in phosphate-buffered saline [PBS]) was performed. The mouse's head, along with the EIB implant, was post-fixed in PFA (4% in PBS) for five days at 4 °C. Subsequently, the EIB was removed, the brain was cut into 50 μm thick sections with a vibrating microtome (Thermo Scientific Microm HM 650 V), and the tetrode traces were identified using a bright field microscope. Tetrode tracks were manually reconstructed from the microscope images using the Amira software (v6.5, Thermo Fisher Scientific, Waltham, MA). Subsequently, all tetrodes located outside of the designated target areas were excluded from further analysis.

### Calcium imaging

**Viral Injection.** For intracranial viral injections, mice were anesthetized with isoflurane (1–2%, Univentor 1200 - anesthesia Unit, Univentor, Zejtun, Malta). Mice were head-fixed in a Kopf Stereotax frame (Model 900, David Kopf Instruments, Tujunga, USA). Deep anesthesia was ensured by the absence of the paw pinch reflex. The head was shaved to expose the skin and disinfected with ethanol swabs. The skin was incised along the rostral-caudal axis of the head, and the surface was cleaned with cotton swabs. An oval-shaped craniotomy was made at (M/L: 1.25, A/P: −1.4 to −2.0, D/V: 3.2) mm relative to bregma using a dental drill (EXL-40, Osada Electric co. ltd.). To achieve optimal expression in POm, we injected the virus solution (AAV-1/2-GCaMP6f, 4.4 ×1012 GC, 400 nl) at two coordinates (M/L: 1.25, A/P: −1.4, D/V: 3.2; M/L: 1.25, A/P: −2.0, D/V: 3.2). A pulled glass injection pipette (Puller: Model P-97, Sutter Instrument Co., Novato, USA; Pipette: Blaubrand intraMark micropipettes, product ID 708707) loaded with the virus was slowly lowered to the injection site and the virus was slowly injected (-100 nl/min) by manually applying air pressure with an attached syringe. The tissue was allowed to rest for 10 min to facilitate diffusion of the virus before the pipette was slowly pulled back up. The skin was sutured back up with a Braun suture kit.

**Table 1 | Parts list**

| Product | Supplier |
|---|---|
| Linear track (82 × 10 cm, wood) | Custom built, Lee Embray, Heidelberg |
| Controlling unit of the behavioral apparatus (Arduino Uno R3 with firmata firmware) | Arduino, assembled by Lee Embray, Heidelberg |
| **Apertures** | |
| Metal wings (height: 205 mm, width: 97 mm, aluminum) | Lee Embray, Heidelberg |
| Controlling unit with stepper motors (https://github.com/GrohLab/A-tactile-discrimination-task-to-study-neuronal-dynamics-in-freely-moving-mice/blob/main/MaterialList/ApertureGate/info.md)[41] | Lee Embray, Heidelberg |
| **Lick ports** adapted from Cornelius Schwarz, Tübingen[74] | |
| Vasofix® Safety Braunüle® FEP G16 | B. Braun SE, Melsungen, Germany |
| Piezo sensor (#285-784) | RS components GmbH, Mörfelden-Walldorf, Germany |
| Analog-to-digital converter (ADC) | Cornelius Schwarz, Tübingen[74] |
| **Sensors and IR illumination** | |
| Light beams (https://github.com/GrohLab/A-tactile-discrimination-task-to-study-neuronal-dynamics-in-freely-moving-mice/blob/main/MaterialList/LightBeams/info.md)[41] | Lee Embray, Heidelberg |
| LED strips for overhead illumination (Synergy 21 LED Flex Strip infrarot IR 12 12 V IP65 Security Line 850 nm, #140455) | ALLNET GmbH, Germering, Germany |
| Backlight LEDs (HT-110IRPJ IR-Emitter 850 nm 140° 1204 SMD) | Harvatek Corporation, Hsinchu, Taiwan |
| Diffuser plate (3D printed with clear FLGPCLO4 and sanded with P600) | Formlabs GmbH, Berlin, Germany |
| Diffuser film, 60% light diffusion (#DSF-10) | Folien Fischer UG, Hamburg, Germany |
| **Intervention drugs** | |
| Isoflurane (Isofluran Baxter) | Baxter Deutschland GmbH, Germany |
| Buprenorphine (Bupresol vet. Multidose 0,3 mg/ml, 10 ml) | CP-Pharma Handelsgesellschaft mbH, Burgdorf, Germany |
| Ketamine (100 mg/ml, 10 ml) | CP-Pharma Handelsgesellschaft mbH, Burgdorf, Germany |
| Xylazine (Xylavet® 20 mg/ml) | CP-Pharma Handelsgesellschaft mbH, Burgdorf, Germany |
| Lidocaine (Xylocaine® 1%) | Aspen Pharma Trading Limited, Ireland |
| ssAAV-1/2-hSyn1-chI-GCaMP6f-WPRE-SV40p(A) | Viral Vector Facility (VVF), Zürich, Switzerland |
| N-Methyl-D-aspartic acid, 98% (#AC329190500) | Thermo Fisher Scientific Inc., MA, USA |
| **Cameras** | |
| 2x High speed cameras (DMK 37BUX287) | The Imaging Source Europe GmbH, Bremen, Germany |
| Overview camera (DMK 37BUX290) | The Imaging Source Europe GmbH, Bremen, Germany |
| **Reward/punishment** | |
| Food dispensers (https://github.com/GrohLab/A-tactile-discrimination-task-to-study-neuronal-dynamics-in-freely-moving-mice/blob/main/MaterialList/FoodDispensers/info.md)[41] | Lee Embray, Heidelberg |
| 2 ml syringes (BD Discardit II) | Becton Dickinson GmbH, Heidelberg, Germany |
| Tubes with Luer connector (#4256000) | B. Braun SE, Melsungen, Germany |
| Hifi integrated amplifier (A-S301) | Yamaha Music Europe GmbH, Rellingen, Germany |
| 2x Speakers (Neo X 3.0 ribbon tweeter) | Fountek Electronics Co., Ltd, Zhejiang, China |
| **EIB** | |
| RHD USB Interface Board (#C3100) | Intan Technologies, CA, USA |
| Samtec Male Connector EIB (ST4-40-1.00-L-D-P-TR) | Samtec Europe GmbH, Germering, Germany |
| Samtec Female Connector Headstage (SS4-40-3.00-L-D-K-TR) | Samtec Europe GmbH, Germering, Germany |
| PCB (EIB and headstage) | Custom built (BRD file: https://github.com/GrohLab/A-tactile-discrimination-task-to-study-neuronal-dynamics-in-freely-moving-mice/blob/main/MaterialList/EIB64-ST4-Multi.brd)[41], printed by Multi-Circuit-Boards Ltd, Munich, Germany |
| Tungsten 99.95% CS Wire for Tetrodes(#100211) | California Fine Wire Company, CA, USA |
| Gold chloride solution for gold plating (200 mg/dL gold chloride in deionized water, HT1004-100ML) | Sigma-Aldrich, Inc., MO, USA |
| Intan Interface 64 Channel Amplifier Chip (RHD2164) | Intan Technologies, CA, USA |
| UV-curable adhesive (3 M™ Filtek™ Supreme XTE) | 3 M Deutschland GmbH, Neuss, Germany |
| Function generator (A365 Stimulus Isolator) | World Precision Instruments, Inc., FL, USA |
| Electric Rotary Joint (AHRJ-OE_2×2_AD_200-0.22_24_USB-C) | Doric Lenses Inc., Quebec City, Canada |
| **Miniscope V4** | |
| Miniscope V4 main body (Excitation, objective, and emission module) | Custom print from Heidelberg precision mechanics workshop/ Open Ephys |
| V4 Base Plate | Custom print from Heidelberg precision mechanics workshop/ Open-Ephys |
| Rigid-Flex PCB | Open-Ephys |

**Table 1 (continued) | Parts list**

| Product | Supplier |
|---|---|
| - Optical Filters -<br>Excitation filter 4 mm x 4 mm, ET47/40x<br>Emission filter 4 mm x 4 mm, ET525/50 m-<br>Dichroic mirror 6 mm x 4 mm, T495lpxr | Chroma/ Open-Ephys |
| - Lenses<br>2x3 mm diameter, 6 mm FL achromat used in the objective module (#45-089)-<br>4 mm diameter, 10 mm FL achromat used in the emission module (#63-691)-<br>Half-ball Lens, 3.0 mm Diameter, N-BK7 (#47-269) | Edmund Optics/ Open-Ephys |
| Coax cable, 36AWG 26/50SPC | Cooner Wire |
| Hirose U.FL Connector, U.FL-PR-SMT2.5-1(10) | Mouser |
| GRIN Lenses, ProView™ Lens Probe, 0.5 mm diameter, ~4.0 mm length | Inscopix |

**Table 2 | Behavioral paradigm: training stages, stimuli, responses, scoring, outcomes and progression criteria**

| Stage name | Stimulus | Response | Scoring (outcome) | Criterion |
|---|---|---|---|---|
| Habituation | 45 mm aperture | Lick | Hit (reward) | ≥ 4 sessions and ≥ 50 successful lick trials |
|  |  | No lick | Miss (no outcome) |  |
| Initial rule | 45 mm aperture | Lick | Hit (reward) | ≥ 4 sessions and ≥ 200 trials above expert-performance threshold (d' = 1.65) |
|  |  | No lick | Miss (no outcome) |  |
|  | 25 mm aperture | Lick | FA (punishment) |  |
|  |  | No lick | CR (no outcome) |  |
| Neutral aperture stage | 45 mm aperture | Lick | Hit (reward) | 8 sessions in total |
|  |  | No lick | Miss (no outcome) |  |
|  | 35 mm aperture | Lick | No score (no outcome) |  |
|  |  | No lick | No score (no outcome) |  |
|  | 25 mm aperture | Lick | FA (punishment) |  |
|  |  | No lick | CR (no outcome) |  |
| Reversed rule | 45 mm aperture | Lick | FA (punishment) | ≥ 4 sessions and ≥ 200 trials above expert-performance threshold (d' = 1.65) |
|  |  | No lick | CR (no outcome) |  |
|  | 25 mm aperture | Lick | Hit (reward) |  |
|  |  | No lick | Miss (no outcome) |  |
| Extinction | 45 mm aperture | Lick | No score (random presentation of reward and punishment) | 8 sessions in total |
|  |  | No lick |  |  |
|  | 25 mm aperture | Lick |  |  |
|  |  | No lick |  |  |

*FA* false alarm, *CR* correct rejection

**Grin-Lens Implantation.** Two to three weeks after the viral injection, the animal was anesthetized and head-fixed on the Stereotax. A small portion of the skin was removed from the head to expose the skull. The connective tissue was removed with cotton swabs. After adjusting the head in the correct 3D plane, a craniotomy was drilled between the previous injections (M/L: 1.25, A/P: −1.7 relative to bregma). A blunt needle was used to make a leading track for the lens. The needle was advanced with a Luigs & Neumann linear actuator by advancing in 300 μm steps and retracting in 200 μm steps at the speed of 30 μm/s to relieve pressure to a final depth of 3000 μm from the dura. The GRIN lens (Inscopix, product code 1050-004627, ProView™ Lens Probe, 0.5 mm diameter, ~4.0 mm length) was held with a 3D printed custom lens holder. The lens was lowered slowly in the same way as the blunt needle, except for the final depth being 3100 μm. The lens was then fixed to the skull with a self-curing resin cement (Super-bond, Sun Medical, Japan).

**Baseplate Implantation.** After the GRIN lens implantation, a baseplate was attached to a miniscope, which was lowered while acquiring images through the GRIN lens. The baseplate was lowered to the focal plane and then fixed to the skull with UV-cured dental cement (Gradia Direct Flo, A3, GC Corporation).

**Sensory deprivation protocols (whisker plucking and lidocaine numbing of the whisker pad)**
For whisker plucking, mice were anesthetized with 2.5% isoflurane and then placed in a stereotaxic alignment system (Kopf Instruments) with a mask providing continuous inhalation of 1–2% isoflurane in medical oxygen. The body temperature was monitored and maintained at 37–39 °C. The mouse's head was tilted to see the whisker pad clearly under the microscope, and lidocaine cream was applied to the whisker pad. The whiskers were then plucked one by one using small cosmetic tweezers. After plucking all the whiskers (on both sides), mice were kept on a heating plate in their preheated cage for a minimum of 30 min to recover, and they were given at least 5 h of rest before their next training session. In a separate sensory deprivation protocol, lidocaine cream was applied topically to the whisker pads during a brief anesthesia with isoflurane approximately 20–30 min in advance of the session. For control, neutral ointment (Bepanthen cream) was

**Table 3 | Tetrode Coordinates**

| Tetrode | A/P | M/L | D/V with respect to dura |
|---|---|---|---|
| BC 1 | −1.0 | 3.25 | −0.4 |
| BC 2 | −1.3 | 3.3 | −0.3 |
| BC 3 | −1.6 | 2.7 | −0.85 |
| BC 4 | −1.3 | 2.6 | −0.85 |
| BC 5 | −1.0 | 2.7 | −0.85 |
| VPM 1 | −1.95 | 1.95 | −2.95 |
| VPM 2 | −1.95 | 1.6 | −2.95 |
| VPM 3 | −1.75 | 1.75 | −3.0 |
| VPM 4 | −1.45 | 1.6 | −2.95 |
| POm 1 | −2.15 | 1.3 | −2.75 |
| POm 2 | −2.3 | 1.5 | −2.75 |
| POm 3 | −2.3 | 1.15 | −2.8 |
| POm 4 | −2.0 | 1.05 | −2.85 |
| ZIv 1 | −2.35 | 2.0 | −3.7 |
| ZIv 2 | −2.6 | 1.75 | −3.7 |
| ZIv 3 | −2.15 | 1.75 | −3.75 |

*BC* barrel cortex, *VPM* ventral posteromedial nucleus, *POm* posterior medial complex, *ZIv* ventral zona incerta

**Table 4 | Coordinates for OB lesion**

| Site | A/P | M/L | D/V with respect to dura |
|---|---|---|---|
| 1 | +5.25 | +/−0.75 | −1.5 |
| 2 | +4.5 | +/−0.75 | −2.0 |
| 3 | +3.75 | +/−0.75 | −2.5 |

administered identically. The whiskers remained intact for this intervention.

**Barrel cortex lesions**
Mice were anesthetized with 2.5% isoflurane and then placed in a stereotaxic alignment system (Kopf Instruments) with a mask providing continuous inhalation of 1–2% isoflurane in medical oxygen. Carprofen (CP-Pharma) was given subcutaneously at a dose of 5 mg/kg, while lidocaine (Xylocaine 1%, Aspen Pharma) was injected beneath the scalp and around the ear bars for local anesthesia. To keep the eyes lubricated during the surgery, Bepanthen ointment (Bayer) was applied. Once tail- and toe-pinch reflexes were confirmed absent, a midline incision was made in the skin, and the periosteum and aponeurotic galea were excised to expose the reference points bregma and lambda. The head was then properly aligned with the stereotaxic frame. Craniotomies were performed using the previously mapped borders of the barrel cortex on the skull as anatomical landmarks to ensure complete exposure of the barrel cortex. Next, a fine glass pipette was carefully inserted to a depth of 1.4 mm at the edge of the craniotomy. It was then moved along the contour 4 to 5 times until the brain tissue began to detach. Finally, the tissue was pulled and excised using thin needle bevels, and the surgical excision was complete when the white matter was visible.

**Olfactory bulb lesions**
Surgical preparations were conducted the same way as in the barrel cortex lesion procedures. A Hamilton syringe (10 µl) was used to inject N-methyl-D-aspartate (NMDA, 20 µg/µl, 1 µl per injection site, Thermo Fisher Scientific Inc., MA, USA) into the OB of the lesion group. Each hemisphere received injections at three sites (Table 4, coordinates relative to bregma). The needle was first lowered to the most ventral

point of the injection, and NMDA was slowly delivered while retracting the syringe up to −0.3 mm below the dura, with a continuous injection rate of 100 nl/min. For control, the same surgical procedure and injection protocol were followed, but with sterile saline solution replacing NMDA.

**Data analysis**
The behavioral data was analyzed using custom code developed in MATLAB R2023a (The MathWorks, Inc.), which is available on: https://github.com/GrohLab/A-tactile-discrimination-task-to-study-neuronal-dynamics-in-freely-moving-mice/tree/main/DataAndScripts[41].

$P \leq 0.05$ were considered significant. Mean values were reported with standard deviations and median values with interquartile range (IQR).

**Statistical analysis of the learning process.** Task performance in each session was measured using the discriminability index d-prime (d'), also known as Fisher discrimination index[62,63]. The d' is a statistical measure to quantify the ability to discriminate between signal and noise and is defined as the difference between the z-scores of the hit rate and the FA rate: $d\prime = z(hit\ rate) - z(false\ alarm\ rate)$. Due to the constraints of the z-score transformation (which cannot handle proportions of 0 or 1), hit rates and FA rates of 0 are typically adjusted to $(\frac{1}{2*n})$ and rates of 1 are adjusted to $(1 - \frac{1}{2*n})$, with $n$ being the number of go- (for hit rate adjustment), or no-go trials (for FA rate adjustment), respectively. A higher d' indicates greater discriminability, meaning the observer can more effectively distinguish signal from noise. A d' of 0 describes a discrimination performance at chance level. An animal was considered to reach expert performance when it crossed a d' of 1.65, corresponding to a one-tailed significance level of α = 0.05.

Success rates for go, no-go trials and all trials were calculated as follows: Hit/(Hit+Miss), CR/(CR + FA) and (Hit+CR)/(Hit+Miss+CR + FA), respectively. The moment of insight was defined as the point where the no-go success rate increases from one session to the other by a factor of 5. To calculate the population moment of insight, each individual moment of insight was interpolated on a common access and then the mean was taken across all mice.

**Extraction of behavioral data from videos.** Spatial information of the mice was extracted from video recordings using DeepLabCut[23]. Following preprocessing with median background image subtraction, the DeepLabCut model was trained on a subset of manually annotated video frames. For the high-speed videos (240 Hz), these markers consisted of 8 data points for two whiskers on each side from whisker base to tip and markers for the mouse's head contours. To validate tracking accuracy, we estimated the mean average Euclidean error (MAE) between manually annotated labels and DeepLabCut predictions. The estimated MAEs were 3.44 pixels (≈0.3 mm) for the training set and 10.39 pixels (≈1.0 mm) for the test set. We further investigated the consecutive coordinate differences of labels with the inbuilt DeepLabCut function. Instances of label jumps typically affected all labels associated with a specific whisker while preserving their relative spatial arrangement, minimizing the potential impact on our whisker trajectory analysis.

In overview videos, spatial positioning and velocity were estimated based on anatomical landmarks on the mouse's body and head. For the positional data analysis (Fig. 5h), the linear track was divided into rectangles measuring 10 mm by 1 mm. The time spent in each area was calculated by counting the number of video frames in which the mouse was present within each specific rectangle.

**Lick events.** Trendlines in Fig. 5e were computed with a best-fit asymptotic function: $y = a + b \cdot e^{(-c \cdot x)}$; where a is the respective median lick latency, b is the initial deviation from the asymptote, and c

is the decay rate of latency reduction across trials. The fitting process used least-squares optimization with a MultiStart approach to avoid local minima.

**Electrophysiological data.** The raw electrophysiological data were spike-sorted with KiloSort 2.0. Subsequently, the sorted clusters were curated automatically using the Ecephys spike sorting algorithm provided by the Allen Brain Observatory (code available at https://github.com/alleninstitute/ecephys_spike_sorting). The algorithm includes noise templates to identify and exclude clusters exhibiting characteristics indicative of noise, such as irregular waveform shapes and interspike interval (ISI) histograms. The following quality metrics were then applied to further exclude any clusters indicative of multi-unit activity: isolation distance (computed from Mahalanobis distance) greater than 15 and ISI violation below 3%, with an ISI threshold of 1.5 milliseconds. The criteria were selected based on previous studies[64–66], where they have been demonstrated to effectively distinguish well-isolated single units from noise and multi-unit activity. The individual units were further classified into fast spiking (FS) and regular spiking (RS) units, putatively representing inhibitory and excitatory units, respectively[67]. The threshold for differentiating RS from FS units was set to a trough-to-peak duration of 350 μs for cortex and zona incerta, and 300 μs for thalamic nuclei. These thresholds align with previous studies[68,69] and consider the fact that extracellular waveforms in the thalamus are shorter than those in the cortex[68]. For further analysis, only RS units were selected for BC, VPM, and POm, while FS units were selected for ZIv.

**Extraction of calcium traces and spatial footprints.** The preprocessing, motion correction, and extraction of the calcium traces and spatial footprints from the videos was done with Minian[70]. All downstream processing was done using custom code developed in MATLAB R2023a (The MathWorks, Inc.).

**Detection of response onset latency.** To determine the onset latency of neuronal responses following whisker contact, we employed a statistical approach based on a Poisson model of baseline firing rates, similar to de Kock et al.[71]. Firing rates were first binned into bins of 1 ms. The baseline firing rate was then estimated as the mean firing rate within a pre-stimulus window (800 ms to 600 ms before stimulus onset). The first bin within a window of 200 ms after touch onset in which the observed firing rate significantly exceeded the baseline firing rate in a Poisson modeled distribution ($p \leq 0.05$) was identified as the response onset latency.

**Touch modulated units.** Calcium transients of the POm units were extracted in a window of 400 ms before and after the whisker touch (see above). The summed calcium activity before and after the touch was compared with a paired two-tailed $t$-test. Whisker touch responsiveness for the extracellular recordings was determined by a statistically significant increase in firing rates (to either of the two aperture states) within a 200 ms response window following aperture touch, compared to a baseline window preceding touch onset. Those units which showed significant differences were identified as touch-modulated units, using a two-sided Wilcoxon signed rank test.

**Decoding of aperture width form unit spike trains.** Aperture width decoding from unit spike trains traces was conducted using the Neural Decoding Toolbox[72]. The dataset was partitioned into a training set (90% of labels) and a test set (10% of labels). Each set contained the spike traces of individual units for a given trial and the corresponding aperture labels. A support vector machine (SVM) classifier was trained using the LIBSVM software package. The classifier was subjected to a 10-fold cross-validation, where it was trained and tested on different data partitions. The mean decoding accuracy was then calculated over a window from trigger onset to 400 ms post-trigger onset.

**Decoding of aperture width from whisker angles.** Aperture width decoding from whisker angles was performed using MATLAB's built-in generalized additive model (GAM) classifier. The dataset was partitioned into a training set (80% of labels) and a test set (20% of labels). Each set included whisker angles recorded 20 frames before and 100 frames after whisker touch, with a sampling rate of 240 fps, along with the corresponding aperture labels. In the shuffled decoding accuracy analysis, the whisker angle traces were randomly allocated to an aperture label. $P$-values were determined using pointwise 95% confidence intervals, which were computed by the MATLAB "rocmetrics" function based on 100 pointwise bootstrap resamples.

**Decoding of aperture width from the calcium imaging data.** Calcium transients were extracted 400 milliseconds (12 frames) before and 400 miliseconds (12 frames) after the whisker touch, and a six-layer ReLU-activated convolutional neural network (CNN) was designed and trained for binary classification of calcium transients. The network was provided with the extracted calcium transients from neurons which showed activity during all of the expert sessions. Feature extraction was performed through two convolutional blocks, each consisting of a $3 \times 3$ convolutional layer with padding applied to maintain the input dimensions, followed by batch normalization to stabilize learning and a ReLU activation function to introduce non-linearity. Dimensionality was reduced through max pooling layers, which downsampled the feature maps by a factor of 2. Once features were extracted, classification was performed by a fully connected layer with two output neurons, corresponding to the two target classes. A softmax layer was applied to convert the outputs into class probabilities, and predictions were finalized by a classification layer. The network was trained using the stochastic gradient descent with momentum (SGDM) optimizer. Training was conducted over 100 epochs, with an initial learning rate of 0.01.

**Analysis of neuronal activity during whisking and quiescence.** Whisking behavior was analyzed using a 100 ms moving window, with a 50 ms margin before and after each time point. Periods of free whisking were defined as those with whisking amplitudes exceeding 5° within the specified time window, whereas quiescent periods were characterized by amplitudes below 3°. Frames associated with aperture touch events or occurring within 100 ms of such events were excluded from analysis to minimize potential confounds introduced by touch-related neuronal responses. Statistical significance of whisking angle tuning was determined using a shuffling-based test. Spike times for each unit were shifted by a random interval (uniformly distributed and less than the total recording duration), wrapping the end of the shifted sequence to the beginning of the original sequence. This process preserved spike counts and firing structure while decoupling spikes from behavior. Shuffling was repeated 100 times for each unit. Differences in observed firing rates exceeding the 95th percentile of the shuffled distribution were deemed significant.

**Analysis of neuronal activity during locomotion and resting periods.** Locomotor activity was quantified using video recordings to extract mouse velocity within a designated region of interest along the linear track. This approach minimized interference from whisker interactions near reward sites or apertures. Velocity estimates were derived using spatial features tracked via DeepLabCut. Locomotion was defined as sustained velocities exceeding 100 mm/s for longer than 200 ms, while resting was characterized by velocities below 10 mm/s persisting for at least 200 ms. Statistical significance was determined using the same shuffling-based method applied in the analysis of neuronal activity during whisking and quiescence.

**Analysis of whisking angle tuning.** Whisking angles were analyzed in accordance with prior studies[32]. Angles were binned into 5° intervals, ranging from 0° (maximal retraction) to 180° (maximal protraction). For each bin, firing rates were computed as the number of spikes divided by the total duration within that bin. To exclude touch-related neuronal activity, spikes occurring during or within 100 ms of aperture touch events were excluded. Spikes for each unit were shuffled in the same manner as described above, for statistical testing. Modulation depth $(r_{max} - r_{min})/r_{mean}$, where $r_{max}$, $r_{min}$, and $r_{mean}$ are the maximum, minimum, and mean firing rates, respectively, was considered significant if it exceeded the 95th percentile of the shuffled distribution.

**Analysis of whisking phase tuning.** Whisker position traces were band-pass filtered between 2 and 30 Hz to isolate whisking cycles. Each cycle was divided into 32 bins spanning 360° (or $2\pi$). Instantaneous whisking phase was extracted using the Hilbert transform and normalized to the range $[0, 2\pi]$. Spikes were assigned to their corresponding phase bins, and firing rates were calculated for each bin. Phase-dependent modulation of neuronal firing was assessed by fitting a sine wave to the firing rates using linear least-squares regression, as outlined in Moore et al.[56]. In brief, the modulation of each unit's activity was quantified by fitting a sine wave with a period of $2\pi$ using standard linear least-squares regression. The sine wave fit is defined as:

$$\lambda(t) = \langle\lambda\rangle + Amp_{tuning} \cdot \cos(\phi(t) - \phi_{preferred}),$$

where $\lambda(t)$ represents the mean firing rate, $Amp_{tuning}$ denotes the amplitude of modulation, $\phi(t)$ the whisking phase at time $t$, and $\phi_{preferred}$ the preferred phase at which the unit's firing is maximally modulated: $\phi_{preferred} = 0$ corresponds to whiskers being fully retracted, and $\phi_{preferred} = \pi$ corresponds to full protraction.

The signal-to-noise ratio (SNR) of the phase tuning was then calculated as:

$SNR = 2 \cdot Amp_{tuning}\sqrt{\frac{T}{\langle\lambda\rangle}}$, where $T$ is the temporal window for a point process with Poisson-distributed spike arrival times, and is estimated from the average whisk cycle to be 111 ms. Statistical significance of modulation was evaluated using a Kuiper test applied to the firing rate distribution across phase bins.

**Analysis of head angle tuning.** Head angles were categorized into 5° bins, ranging from −90° (head oriented to the left) to 90° (head oriented to the right). For each bin, firing rates were calculated as the number of spikes divided by the total duration spent within the bin. Modulation indices were computed similarly to whisking angle analyses, and significance was assessed using the same shuffling-based procedure. Both allocentric head angle (relative to the environment, measured via high-speed video recordings) and egocentric head angle (relative to the body axis, derived from overview camera recordings) were analyzed to determine whether neuronal responses differed significantly between these reference frames.

**Analysis of spatial tuning.** To assess spatial tuning, firing rates were calculated in 100 ms time bins and averaged based on maze position, which was divided into 20 mm spatial bins. Following previous studies the firing rates were modeled using a Gaussian function[73], incorporating an additional linear term to account for speed effects. Units were classified as spatially tuned if the amplitude of the Gaussian fit was significantly different from zero (outside the 95% confidence interval) and the linear speed coefficient was non-significant (within the 95% confidence interval), suggesting a location-specific neuronal response independent of locomotor velocity.

## Reporting summary

Further information on research design is available in the Nature Portfolio Reporting Summary linked to this article.

## Data availability

All the data generated in this study have been deposited in the Zenodo database under accession code https://doi.org/10.5281/zenodo.15051370[41]. The data underlying the figure panels generated in this study are provided in the Supplementary Information/Source Data file. Source data are provided with this paper.

## Code availability

Code necessary to reproduce the Matlab-generated figures in this study are provided and maintained at: https://doi.org/10.5281/zenodo.15051370[41] and https://github.com/GrohLab/A-tactile-discrimination-task-to-study-neuronal-dynamics-in-freely-moving-mice/tree/main/DataAndScripts

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

## Acknowledgements

We thank Katharina Ziegler for helpful comments on the manuscript. We thank Wolfgang Kelsch and Max Scheller for teaching us the electronic interface board approach as well as Cornelius Schwarz for providing us with the lick ports. This work was supported by the German Research Foundation (DFG Grants GR3757/4-1 to AG and KU 1983/8-1 to TK). We acknowledge the data storage service SDS@hd and high-performance computing initiative bwHPC, supported by the Ministry of Science, Research and the Arts Baden-Württemberg (SDS@hd and bwHPC) and the German Research Foundation (DFG) through grants INST 35/1597-1 FUGG (bwHPC) and INST 35/1503-1 FUGG (SDS@hd). For the publication fee we acknowledge financial support by Heidelberg University. The funders had no role in study design, data collection and analysis, decision to publish, or preparation of the manuscript.

## Author contributions

F.H., J.T., N.S., M.C., T.K., and A.G. designed the experiments. L.E., F.H., N.S., M.K., and M.C. constructed the setup under supervision of A.G.; M.K. developed Syntalos under supervision of M.B.; J.T., F.H., N.S., M.H.B.-G., J.M.-C., and A.A. collected and analyzed the data under supervision of A.G.; J.T., F.H., N.S., A.A., and A.G. wrote the paper.

## Funding

## Competing interests

The authors declare no competing interests.
