## [Transparent Peer Review file · Nature Communications]

A tactile discrimination task to study neuronal dynamics in freely-moving mice

Corresponding Author: Dr Alexander Groh

Version 0:

Reviewer comments:

Reviewer #1

(Remarks to the Author)

The study introduces a novel tactile discrimination task designed for freely moving mice, with a primary focus on simultaneous recording of head, whisker and neuronal dynamics during sensory processing and learning. By integrating electrophysiology and calcium imaging, the researchers were able to record neuronal activity across multiple brain regions as the mice discriminated between apertures of varying widths using their whiskers. The task revealed that key brain areas, including the barrel cortex, VPM, POM and zona incerta, exhibited significant modulation in neuronal firing in response to whisker-object interactions. These findings provide important novel insights into how thalamo-cortical circuits encode and process tactile information in real-life conditions. It might be in place here to compare the current findings to Oram et al's (Nat Comm earlier this year).

Neuronal recordings demonstrated that over 60% of neurons in the BC and ZI, and around 50% in the VPM and Pom, showed changes in firing rates following whisker touches. The use of calcium imaging in the Pom revealed a discrepancy between spike recording and calcium imaging regarding the fraction of touch-activated neurons, possibly explained by different temporal sensitivities and potential sampling biases – these are important differences that should be taken into account in experiments using only one of the methods. In general, these results provide insights into the involvement of these thalamic nuclei in tactile perception. The behavioral aspect of the study demonstrated that mice quickly learned to discriminate apertures and adapt to rule changes, such as reward-punishment reversals.

Overall this is an impressive tour-de-force integrated into a highly promising system for studying the neuronal basis of behavior and perception in real-life conditions, and presenting the profile of thalamo-cortical activity and information content during a specific tactile task.

Major comments

1. I suggest emphasizing more the thalamo-cortical results. As these results contain novel important aspects, they deserve to be (in my opinion) at the front when presenting the paper (title and abstract). But this is of course only a recommendation – the authors should decide.
2. Several additional doable analyses would complete the picture here: 1. The modulation of neuronal activity in the different brain locations during free-air whisking, and, if possible, also the information carried by these modulations. 2. The dynamics of extinction (in addition to Fig. 2E).
3. Were there insight-like moments in mice behavior during learning? This should be analyzed and reported as both results – insight-driven learning or insight-less learning – are of value to the community.
4. Coding. It is one thing to report that neuronal activity contains information about the external world, and another to show how the external world is coded by this activity. The wonderful system the authors developed allows them to go beyond the first question, and address the second, important, question. Here it would be also interesting to compare rate and temporal codes, codes that were observed earlier to differ in their information content regarding different objects' features (see eg Knutsen-Ahissar 2009).
5. Fig. 6D – it seems that the effect of aperture on whisker-angle might not be a consequence of motor modulation by rats, but rather technically confounded by the rats needing to push their head into the aperture. Please refer to this point in the paper.
6. Whisker tracking is done on two whiskers per side. It is not stated whether these are always the same whiskers – please add it. In a more general sense, the authors should show – in any way they choose to - that their sampling, accuracy and resolution of whisker measurements is appropriate.
7. The use of d' (sensitivity analysis) should preferably be accompanied by some more raw data about % success for narrow and wide apertures. It is unclear how much systematic bias plays a role in the data when only d' is presented.

Minor comments

1. Several places are missing statistical analyses to back-up claims: e.g. – Fig 3A, no p-value is given for the claim that 2mm aperture is performed at chance level (although this is visible from the figure); Fig. 4A,B,C – no comparisons of groups (i.e. go and no-go trials) is made as far as I can tell, but the text discusses their differences. For Fig 4C a trend-line would be a nice addition; Fig 6C – statistically compare the two timeseries somehow, also add error-bars.
2. When referring to the coordinates of the rodent brain, the authors use the terms “anterior” and “posterior”. It seems that the terms “rostral” and “caudal” better fit the customary use for animals with a linear nervous system.
3. The authors “revealed a rhythmic pattern of whisking at 18.05 ± 3.19 Hz”, mentioning that it is consistent with previously described by Mitchinson et al. (2011) whisking frequencies in freely moving mice. In fact, Mitchinson’s mice were whisking with significantly lower frequency, only 11.35 ± 0.95 Hz (see Table 1 in Mitchinson et al., 2011).
4. “Material & Methods”, subsection “Histology and identification of tetrode positions”: it should be indicated at what temperature the post-fixation was carried out for five days.
5. “The ability of mice to relearn repeated rules reversals more efficiently suggests the involvement of higher cognitive processes.” – I don’t think this can be deduced directly from these results.

Comments to figures

1. In the legend to Figure 2: a) “horizontal line” should be “horizontal dashed line”; b) “initial rule: blue and reversed rule: orange;” can be removed since this is written on the picture itself.
2. In the legend to Figure 3: a) add comma after “6 mm”; b) “Horizontal lines” should be “Horizontal dashed lines”; c) “(blue)” and “(orange)” can be removed since this is written on the picture itself.
3. In the legend to Figure 4: “(initial learning, neutral aperture stage, and reversed rule stage, respectively)” can be removed since this is written on the picture itself.

(Remarks on code availability)

Reviewer #2

(Remarks to the Author)

This study describes learning and performance of an aperture discrimination task for freely moving mice, which the animals solve by using their whiskers. The authors demonstrate that they can decode aperture information from neural activity in different brain regions during the task.

Despite its technical soundness, this study neither describe significant technical advance nor does it provide new biological results. Its novelty is therefore far below the usual high bar of Nature Communications. In addition, the results described could have been analysed much more in depth. It therefore does not provide much useful information for the community.

Novelty issues:

Technically it is not novel to record from freely moving rodents during a tactile task. Recording from freely moving mice is also not novel at all.

1/ Freely moving tactile tasks have been developed by many groups for over 10 years. Here is an example study from 2011 <https://www.pnas.org/doi/10.1073/pnas.1116726109>

2/ Reversal learning behavior is not new. The observation that reversal learning can take more time than initial learning (not systematic) is widely reported across a large number of tasks both freely moving and head fixed. It is a widely used test of cognitive flexibility. See this review <https://www.ncbi.nlm.nih.gov/pmc/articles/PMC5018909/>. Reversal learning is not new for a tactile task either <https://www.nature.com/articles/s41586-020-2704-z>.

3/ Recordings in freely moving behaving rodents during a tactile discrimination task have also been performed in the past. For example by Nikbaht et al. 10.1016/j.neuron.2018.01.003. This study shows that stimulus classification can be decoded from brain activity. Hence, decoding stimuli from brain activity of a freely moving animal is not new either. It has been widely done in head-fixed mice as well, the authors here do not bring any results that differs from previous knowledge.

4/ Electrophysiological recordings in freely behaving animals are common. Here is a 2009 review by the Brecht lab 10.1016/j.conb.2009.08.005 about even more challenging approaches to do electrophysiology in freely behaving rodents. The miniscope used by the authors is not novel all. It has been used in 100s of studies in different behavioral context.

Limits of the analysis.

1/ There is no assessment of whether the task is barrel cortex dependent. This is a major issue as some tasks are and other not (Hong et al, 2018; Rodgers et al. 2021; Harrell et al. 2021). The value of the decoding analysis is diminished by the lack of casual evaluation.

2/ The observation that you can decode gap aperture from whisker movement even with lidocaine in the pad is trivial. It is also trivial that one can decode from the tactile system. It would have been interesting to evidence the coding schemes at play i.e. what whisker movement triggers which type of activity.

3/ There is no use of the imaging data and no comparison between decoding with electrophysiology and imaging.

4/ There is no analysis of the evolution of neural activity with learning (has been done in the past in tactile tasks with precise measures under head fixation).

(Remarks on code availability)

Data and code are available on GitHub not on a proper repository

Reviewer #3

(Remarks to the Author)

The authors present a whisker-dependent sensory discrimination task in which freely moving mice are trained to distinguish aperture widths. The mice convey their percept by collecting condensed milk rewards at lick ports (on either end of a linear track) for some aperture widths while avoiding the lick ports for other widths to elude noise punishments. This task is adapted from a classic study (Krupa et al. 2004) in which freely moving rats were trained to discriminate apertures, and the main departure with respect to the current study is that the rats indicated the perceived aperture (narrow vs wide) by choosing the correct side in a two-port left/right apparatus. The establishment of this task in mice is of value to the scientific community because of the vast toolbox available to unravel the neuronal circuit basis for sensory perception, but the authors do not go as far as to take advantage of this toolbox. In its current form, the manuscript reads more like a methods paper than a research article, so the authors will need to make significant improvements to publish this as such.

In the introduction, the central argument for why this task is an important development is that unlike the popular head-fixed paradigms for studying whisker-based touch sensation in mice, their freely moving paradigm allows "active exploration of the rodent's surroundings". This argument is under-developed. Even in head-fixed conditions, mice can and do move their whiskers which in many studies has been considered "active sensing" (see the work of the Kleinfeld, Ahissar, Peterson, and Feldman labs). In head-fixed virtual reality, rodents can also run and use their whiskers actively (see work from the Helmchen lab). Further, freely moving rodents can use their whiskers passively by holding them still and running along a wall or moving their head with fixed whiskers, so the authors need to be more precise about what they mean by active exploration. In my opinion, the head movements are the big difference in non-head-fixed conditions, so a more complete introduction needs to highlight the importance of head movements in whisker-based sensation and provide evidence from the literature to support (see work from the Diamond lab) and justify the precise advantages of their configuration.

The characterization of the task performance and aperture sensing capabilities of the mice (Figs 1-3) is adequate and the authors show that it depends on intact whiskers at the periphery and not on visual cues.

The analysis of the adaptation of motor behavior during learning (Fig. 4) is insufficient. Panels A, B, and C all focus on licking. In head-fixed go-nogo paradigms, indeed licking is the only "decision" variable, but here, it is apparent from their supplementary videos that in their freely moving configuration, advancing through the aperture towards the lick port is the first decision-revealing movement. The mouse never crosses the gap between the no-go aperture and the lick port in the Correct Rejection trial video and it approaches the lick port and licks in the Hit trial video. They need to characterize this approach/stopping behavior, including head movement and orientations, as well as the whisker/head movements used to sample the aperture throughout training. As eluded to in the commentary above for the introduction, these are the elements of the behavior that are unique in their configuration compared to head-fixed go-nogo and will be of interest to the field. The only non-licking motor behavior presented is running along the track, which reflects task engagement but not the skill of performing the task.

The electrophysiological recordings and imaging data are not adequately analyzed (Figs 5-6). Indeed, it is important to be able to carry out these measurements during the task performance, and it is impressive to record from many regions simultaneously, but classifying responsive percentages of the populations in their respective regions alone is not enough. Here are questions that can and should be answered:

- 1.) How is aperture width encoded by single cells and the population? Are there selective single cells? Is this different across the different brain regions?
- 2.) What other task-related parameters (licking, head movements, stopping, running, whisking, bilateral touch, etc.) are encoded and is this different in the different regions?
- 3.) For the classifier analysis, it is not clear what is happening in the 400 ms window chosen (400 ms after the trigger is crossed). Are the mice only whisking at the aperture during this window or are they already advancing to the reward? What about the temporal dynamics of the classifier performance? How does classification accuracy build up and does it do so with a certain relationship to decision? This can of course be checked in the Go trials only with decision being the time that the mouse advances its head beyond the aperture to collect reward.

Beyond what is reported, I think there are two important dimensions that should be investigated to make the impact of the report sufficient for publication in Nature Communications as a research article:

- 1.) Does task performance require barrel cortex? Lesions and/or muscimol experiments could be designed or optogenetics if possible.
- 2.) Is bilateral whisker sensing required? The authors could perform the whisker plucking or lidocaine treatments to one side and see how this affects task performance and whisking/head movement strategy. If bilateral whisker sensing is needed, discuss how and where could these bilateral signals could be integrated.

(Remarks on code availability)

I did not assess the code.

Version 1:

Reviewer comments:

Reviewer #1

(Remarks to the Author)

The authors addressed all my points adequately. I have no additional comments. Congratulations on an impressive paper!

(Remarks on code availability)

Reviewer #2

(Remarks to the Author)

My main point from the previous round of review was that there is neither technical nor biological novelty in the results reported in the study. The technical breakthrough, if any, is mainly to have successfully put different established techniques together. This is certainly a lot of carefully done work but, in the end, what is really the added value for the reader beyond the fact that putting these techniques together is feasible? The authors argue that it is the first time this type of behavioral task was successfully implemented in freely moving mice. I agree with them. Yet I maintain my assessment that this is a nice result but a quite specialized result and not necessarily at the level of novelty expected for a Nature Communications paper. Many tasks published in this journal are novel but are accompanied with novel biological results, or a new technique. The authors argue that their task is better documented than in the past or that it integrates multiple recording modalities. These important aspects are expected in high quality studies reporting new discoveries nowadays and do not replace a biological result.

The authors must be commended for the additional data provided, in particular for lesion data. They clearly demonstrate that barrel cortex eases task learning and that the task is still feasible without barrel cortex. This is important data for future users and for understanding the specific role of barrel cortex in task performance. It also shows that this is just a sensory discrimination task and that it can be solved even if not perfectly without cortical sensory processing. Hence cortex is partially dispensable/necessary. This should be reported in the abstract.

The authors claim in their abstract that this task is a platform to study cognitive aspects. There is in fact not much in the results corroborating this view. The task is essentially sensory, only half cortex dependent, and the reversal learning, even when repeated, does not accelerate learning in comparison with the initial learning. So, there is really nothing cognitive in this task. This should be removed from the abstract.

The authors have more thoroughly analyzed the representation of touch, in multiple regions of the tactile system, yet they provide no evidence of a particular progression of these representations that could explain why barrel cortex is needed to learn faster the task. The results are not mentioned in the abstract showing their lack of scientific relevance.

The observation of sensitivity to diverse behavioral parameters is important although not surprising given the large amount of literature in the visual (Carandini lab, Striker lab, and many other) or auditory (Kuchibhotla lab, Froemke lab, King lab) system describing this. This is of course the first time this is extensively documented in a freely moving tactile task in mice. Maybe an important confirmation but not surprising in any way.

Overall, I maintain my statement. A solid work with very little novelty.

(Remarks on code availability)

Reviewer #3

(Remarks to the Author)

I would like to commend the authors for having addressed all of the comments. They have done a thorough rebuttal and greatly enhanced the scope and novelty of the study, even sometimes going beyond what was asked of them. The work is complete and suitable for publication.

I would suggest one last read through as there are some small typos and wording issues. One example in line 279 :

"The complete removal of whiskers, deteriorated the performance of expert animals below performance threshold (Fig. 4a). In contrast, animals continued to perform at expert level with IR backlights and IR overhead LEDs turned off, demonstrating that visual cues were not used to solve the task (Fig. S4b)."

This second sentence should explicitly indicate that those animals are whisker-intact. It reads as if those tests were done on the whisker-removed animals mentioned in the sentence before.

(Remarks on code availability)

I did not review the code.

REVIEWER COMMENTS

Reviewer #1 (Remarks to the Author):

The study introduces a novel tactile discrimination task designed for freely moving mice, with a primary focus on simultaneous recording of head, whisker and neuronal dynamics during sensory processing and learning. By integrating electrophysiology and calcium imaging, the researchers were able to record neuronal activity across multiple brain regions as the mice discriminated between apertures of varying widths using their whiskers. The task revealed that key brain areas, including the barrel cortex, VPM, POM and zona incerta, exhibited significant modulation in neuronal firing in response to whisker-object interactions. These findings provide important novel insights into how thalamo-cortical circuits encode and process tactile information in real-life conditions. It might be in place here to compare the current findings to Oram et al's (Nat Comm earlier this year).

Neuronal recordings demonstrated that over 60% of neurons in the BC and ZI, and around 50% in the VPM and Pom, showed changes in firing rates following whisker touches. The use of calcium imaging in the Pom revealed a discrepancy between spike recording and calcium imaging regarding the fraction of touch-activated neurons, possibly explained by different temporal sensitivities and potential sampling biases – these are important differences that should be taken into account in experiments using only one of the methods. In general, these results provide insights into the involvement of these thalamic nuclei in tactile perception. The behavioral aspect of the study demonstrated that mice quickly learned to discriminate apertures and adapt to rule changes, such as reward-punishment reversals.

Overall this is an impressive tour-de-force integrated into a highly promising system for studying the neuronal basis of behavior and perception in real-life conditions, and presenting the profile of thalamo-cortical activity and information content during a specific tactile task.

Major comments

Response: Thank you for your in-depth and scholarly review and the helpful suggestions to improve our paper.

1. I suggest emphasizing more the thalamo-cortical results. As these results contain novel important aspects, they deserve to be (in my opinion) at the front when presenting the paper (title and abstract). But this is of course only a recommendation – the authors should decide.

Response: Following the reviewer's recommendation, we have updated the abstract to place greater emphasis on the thalamo-cortical results, ensuring that their significance is clearly conveyed.

2. Several additional doable analyses would complete the picture here: 1. The modulation of neuronal activity in the different brain locations during free-air whisking, and, if possible, also the information carried by these modulations. 2. The dynamics of extinction (in addition to Fig. 2E).

Response: In regards to point 2.1: we added new analyses of neural encoding of whisker and head kinematics during active sensory exploration in the different brain locations. Specifically, we analyzed the modulation of firing rates by whisking, whisker angle, whisking phase and locomotion (new Figs. 5 and 6). We find varying proportions of units in all four recorded areas that are modulated by these behavioral parameters. Consistent with recent studies in head-fixed awake (Sumser, Isaías-Camacho, Mease, & Groh, 2024) and freely moving mice (Oram, Tenzer, Saraf-Sinik, Yizhar, & Ahissar, 2024), the proportions of whisker angle- and whisking phase-tuned units were comparable in VPM and POM. We included these observations in the paragraph starting at line 423.

In regards to the dynamics of extinction (point 2.2), we added a new analysis (new Fig. S1g) showing the performance- and lick-rate progression over the extinction sessions. The plot illustrates the instantaneous drop of d' to around zero as a result of the design of the experiment: since rewards and punishments, as well as the apertures are randomized, the mice cannot make more correct choices than incorrect choices. The lick rates for the previously punished aperture (wide) reached 100% within the first 3 sessions, demonstrating that mice seek rewards, regardless of the aperture states. This emphasizes that the association between the stimulus and outcomes is successfully broken during the extinction phase and that mice no longer follow the previously learned rule. In other words the aperture states lose their meaning for the mouse behavior. We made this more clear now in the paragraph starting at line 156.

3. Were there insight-like moments in mice behavior during learning? This should be analyzed and reported as both results – insight-driven learning or insight-less learning – are of value to the community.

Response: We addressed this question in a new analysis of the success rates for go and no-go trials using definitions for insight-like learning established in earlier work (Rosenberg, Zhang, Perona, & Meister, 2021). This analysis revealed a sudden performance jump at around 60% of

the stage progression (new Fig. 2b) at which animals increased their success rate in no-go trials by a factor of five. This effect was also observed in the reversed stage (new Fig. S1d). We conclude that this discrimination task involves fast learning with insight-like moments and added this observation in the paragraph starting at line 143.

4. Coding. It is one thing to report that neuronal activity contains information about the external world, and another to show how the external world is coded by this activity. The wonderful system the authors developed allows them to go beyond the first question, and address the second, important, question. Here it would be also interesting to compare rate and temporal codes, codes that were observed earlier to differ in their information content regarding different objects' features (see eg Knutsen-Ahissar 2009).

Response: For the encoding of behavioral states we would like to refer to the new analysis of whisking parameters and locomotion tuning (new Figs. 5 and 6). In regards to encoding of external world parameters, we performed new analyses on putative spatial tuning (new Fig. 5i, j) as well as head-direction tuning (egocentric and allocentric, new Fig. 6f). In agreement with a recent study (Oram et al., 2024), we find that BC, VPM, POm and ZIv carry information about head kinematics. The proportions of egocentric head angle tuned units were comparable in all four areas. Allocentric head-angle tuned unit proportions were similar to egocentric, with the exception of ZIv, which showed a smaller proportion of egocentrically tuned units. Interestingly, we find a small fraction of around 5% of the units being "putative spatially tuned units". We included these observations in the new paragraphs starting at lines 375 and 442.

In regards to the question about rate coding and temporal coding, we would like to refer to the new analysis of whisker angle and whisker phase (new Fig. 6), which can be considered as examples for rate coding. In addition, we also found evidence for temporal coding. During learning, populations of neurons emerged across the recorded cortical, thalamic, and extrathalamic regions, encoding the aperture widths through the presence or absence of bursts. These burst-coding neurons (BCNs) increased in number with task acquisition and BCN burstiness scaled with stimulus valence rather than physical stimulus identity. BCNs dynamically tracked stimulus-reward contingencies across multiple rule reversals, and burst coding

dissipated during the extinction stage. In the context of coding schemes, burst coding may be viewed as a temporal code that scales with stimulus valence. Since this burst coding observation would exceed the scope of this current study, we would like to refer to the preprint of a parallel study: <https://doi.org/10.1101/2025.03.11.642368>

5. Fig. 6D – it seems that the effect of aperture on whisker-angle might not be a consequence of motor modulation by rats, but rather technically confounded by the rats needing to push their head into the aperture. Please refer to this point in the paper.

Response: The reviewer is correct in that the whiskers are more deflected by the aperture during lick port approaches in narrow trials compared to wide trials, as shown in Fig. 6c (now Fig. S4g). As the reviewer points out, this is due to a passive deflection of the whiskers as mice pass through the apertures. Indeed this difference in deflection likely contributes to the decoding of aperture states from whisker kinematics. We clarify this point starting at line 309.

6. Whisker tracking is done on two whiskers per side. It is not stated whether these are always the same whiskers – please add it. In a more general sense, the authors should show – in any way they choose to - that their sampling, accuracy and resolution of whisker measurements is appropriate.

Response: The reviewer is correct, that two whiskers per side were tracked.

Accuracy and resolution: We estimated the DeepLabCut tracking accuracy by calculating the mean average Euclidean error (MAE, proportional to the average root mean square error, RMSE), between the manually annotated labels and the predictions generated by DeepLabCut. Estimated MAEs were 3.44 pixels (\approx 0.3 mm) for the training set and 10.39 pixels (\approx 1.0 mm) for the test set. The below image illustrates a typical test frame with ground truth labels (+) and predictions (dots).

We further investigated the consecutive coordinate differences of labels with the inbuilt DeepLabCut function. Instances of label jumps typically affected all labels associated with a specific whisker, preserving the relative spatial arrangement between labels. Thus, we cannot rule out that the tracking occasionally jumped between whiskers. For our whisking frequency analysis we averaged the whisker angle trajectories across tracked whiskers. Since rhythmic whisking has been reported to be generally synchronized across whiskers (Ahissar & Knutsen, 2008; Sachdev, Berg, Champney, Kleinfeld, & Ebner, 2003; Voigts, Sakmann, & Celikel, 2008), we consider the potential impact of annotation inaccuracies in our estimates to be minimal. We now refer to this point in the revised paper starting at line 803.

Sampling rate: The tracking in our setup uses a video frame rate of 240 Hz. Assuming that whisking frequencies in rats and mice may vary up to 20 Hz during behaviorally salient tasks (Voigts et al., 2008), the minimum sampling frequency required to capture these dynamics would be 40 Hz (Nyquist–Shannon sampling theorem). With a frame interval of approximately 4 ms, this high temporal resolution furthermore ensures relatively precise alignment with neural activity. The temporal accuracy for the touch times is however somewhat limited by the fact that the mice touch the apertures not always with the same whisker in each trial and the first whisker to touch the aperture may not be the tracked one. Nevertheless, using this approach yielded realistic neuronal response latencies (see spike raster plots and new latency analyses in Fig. 3). In conclusion, the sampling rate and tracking accuracy is sufficient for the analysis of whisking frequency and touch times, while the touch time accuracy is somewhat limited by natural variation of whisker-aperture interactions and the frame rate. We expanded on these points in several places of the paper (methods, results, discussion).

7. The use of d' (sensitivity analysis) should preferably be accompanied by some more raw data about % success for narrow and wide apertures. It is unclear how much systematic bias plays a role in the data when only d' is presented.

Response: As suggested, we added new analyses of the success rates. Firstly, we added the overall success rate for initial and reversal learning (new Fig. S1b). Secondly, we included an analysis of the success rates split for Go and No-Go trials (new Fig. 2b and S1d). We conclude that these success rate measures track well with the d' measures (for a direct comparison see new Fig. S1a&b).

Minor comments

1. Several places are missing statistical analyses to back-up claims: e.g. – Fig 3A, no p-value is given for the claim that 2mm aperture is performed at chance level (although this is visible from the figure); Fig. 4A,B,C – no comparisons of groups (i.e. go and no-go trials) is made as far as I can tell, but the text discusses their differences. For Fig 4C a trend-line would be a nice addition; Fig 6C – statistically compare the two time series somehow, also add error-bars.

Response: As suggested, we added the following statistical tests to support the mentioned statements:

- Fig. 3A (now Fig. 2f): one sample t-tests, showing that the 2 mm aperture is performed at chance level ($p = 0.0681$).
- Fig. 4A (now Fig. 2a): Welch's t-test, showing that lick rates for Go and No-Go trials diverge at around 60-70% of the stage progression ($p \leq 0.05$) and lick rates for the neutral aperture are different throughout the entire stage ($p \leq 0.05$).
- We clarified that Fig. 4B (now Fig. 5d) is example data, while population and stats are presented in Fig. 4C (now Fig. 5e).
- Fig. 4C (now Fig. 5e) was split into go and no-go trials, stats and trend lines were added.
- Fig 6C (now Fig. S4g): A Welch's t-test was performed on the two time series using 50 ms time windows. To correct for multiple comparisons, Bonferroni correction was applied. Significant time windows are indicated in the figure by a dashed horizontal line.

2. When referring to the coordinates of the rodent brain, the authors use the terms “anterior” and “posterior”. It seems that the terms “rostral” and “caudal” better fit the customary use for animals with a linear nervous system.

Response: Done

3. The authors “revealed a rhythmic pattern of whisking at 18.05 ± 3.19 Hz”, mentioning that it is consistent with previously described by Mitchinson et al. (2011) whisking frequencies in freely moving mice. In fact, Mitchinson’s mice were whisking with significantly lower frequency, only 11.35 ± 0.95 Hz (see Table 1 in Mitchinson et al., 2011).

Response: In freely moving mice and rats, whisking frequencies have been reported to vary from 11.35 ± 0.95 Hz during locomotion (Mitchinson et al., 2011) to 15–20 Hz during gap crossing tasks (Voigts et al., 2008), indicating that whisking seems to be influenced by motivational and behavioral factors. We reanalyzed the whisking frequency with a larger data set and matched our filtering parameters to the study by Mitchinson et al. 2011 (2 - 30 Hz bandpass). Based on this analysis we estimate a whisking frequency of 14.5 ± 3.1 Hz ($n = 6$ mice), which falls within the range of previously reported values. It is important to note, that in the current setup, these whisking parameters are measured from the high-speed camera videos capturing the aperture / reward zones (see Supplementary videos 1,2). It is thus conceivable that motivational factors such as increased arousal or reward anticipation upon entering the reward zones contribute to the modulation of whisking frequency.

4. “Material & Methods”, subsection “Histology and identification of tetrode positions”: it should be indicated at what temperature the post-fixation was carried out for five days.

Response: We added this information: “The mouse's head, along with the EIB implant, was post-fixed in PFA (4% in PBS) for five days at 4°C.”

5. “The ability of mice to relearn repeated rules reversals more efficiently suggests the involvement of higher cognitive processes.” – I don’t think this can be deduced directly from these results.

Response: We agree and removed this statement.

Comments to figures

1. In the legend to Figure 2: a) “horizontal line” should be “horizontal dashed line”; b) “initial rule: blue and reversed rule: orange;” can be removed since this is written on the picture itself.

Response: Done

2. In the legend to Figure 3: a) add comma after “6 mm”; b) “Horizontal lines” should be “Horizontal dashed lines”; c) “(blue)” and “(orange)” can be removed since this is written on the picture itself.

Response: Done

3. In the legend to Figure 4: “(initial learning, neutral aperture stage, and reversed rule stage, respectively)” can be removed since this is written on the picture itself.

Response: Done

Reviewer #2 (Remarks to the Author):

This study describes learning and performance of an aperture discrimination task for freely moving mice, which the animals solve by using their whiskers. The authors demonstrate that they can decode aperture information from neural activity in different brain regions during the task.

Despite its technical soundness, this study neither describe significant technical advance nor does it provide new biological results. Its novelty is therefore far below the usual high bar of Nature Communications. In addition, the results described could have been analysed much more in depth. It therefore does not provide much useful information for the community.

Novelty issues:

Technically it is not novel to record from freely moving rodents during a tactile task. Recording from freely moving mice is also not novel at all.

1/ Freely moving tactile tasks have been developed by many groups for over 10 years. Here is an example study from 2011 <https://www.pnas.org/doi/10.1073/pnas.1116726109>

Response: We thank the reviewer for the careful review of our manuscript and for appreciating the technical soundness of our study. We understand that the main critique concerns the perceived lack of novelty and biological insights. Motivated by this comment, we substantially increased our analyses of the behavior with several new causal interventions as well as new analyses on neuronal coding of internal and external parameters. The new biological insights include insight-like learning (new Fig. 2b), the necessity of barrel-cortex and bilateral whisker input for normal learning and task execution (new Fig. 4c-e), response-type classification in the different brain areas from electrophysiology and calcium imaging (new Fig. 3e), characterization of turning behavior as a decision variable (new Fig. 5a-c), neuronal encoding of whisker-, head kinematics as well as locomotion and spatial tuning (new Figs. 5i-k, 6).

We further addressed the perceived lack of novelty in the revised version of our manuscript based on the following points:

1. Paradigm for Freely Moving Mice

While the relevant prior studies, including those cited by the reviewer (Nikbakht, Tafreshiha, Zoccolan, & Diamond, 2018) as well as (David J. Krupa, Wiest, Shuler, Laubach, & Nicolelis, 2004; D. J. Krupa, Matell, Brisben, Oliveira, & Nicolelis, 2001), employed tactile discrimination tasks combined with recordings, these were exclusively conducted in rats. To our knowledge, no tactile discrimination task for freely moving mice has been combined with high-resolution neural recordings. Mice offer distinct advantages due to the extensive genetic toolkits available, allowing novel and integrative approaches to studying sensorimotor processing.

2. Integration of Multiple Recording Modalities

Unlike prior studies in rats, our paradigm integrates calcium imaging and electrophysiological recordings within the same experimental framework. This combination provides a more comprehensive view of neural activity across different spatiotemporal scales. In the revised version we now present direct comparisons between imaging and electrophysiological results (see largely overhauled Fig. 3).

3. Detailed Documentation for Reproducibility

A common limitation in existing studies is the lack of detailed information for reproducing experimental setups. In contrast, we provide comprehensive technical documentation, including setup schematics, software code, and step-by-step protocols. This ensures that our approach can be readily adopted and adapted by other researchers.

4. Ethologically Relevant Behavioral Setting

The majority of current whisker-related studies are performed in head-fixed animals, which constrain behavioral richness and naturalistic interactions. Our study demonstrates that tactile learning can be studied in a more ecologically valid, freely moving paradigm, while still achieving high-resolution cellular recordings. This shift toward more naturalistic settings aligns with growing recognition of the need for ethological validity in neuroscience (Datta, Anderson, Branson, Perona, & Leifer, 2019; Krakauer, Ghazanfar, Gomez-Marin, MacIver, & Poeppel, 2017; Miller et al., 2022). We provide concrete examples for harnessing the advantages of the unrestrained over the restrained paradigms. For instance, in the popular head-fixed paradigms, licking is the only decision variable. In a new analysis we demonstrate that decision variables can be obtained from non-licking behavior, such as the turning behavior (new Fig. 5a-c). Another significant limitation of current restrained configurations is that rodents can generally lick all the time, including during stimulus presentation, for recent examples see (Moberg et al., 2025; Petty & Bruno, 2024). This makes it difficult to directly disambiguate sensory-evoked from lick-evoked activity. Our freely-moving paradigm enforces the separation between the sensory sampling phase and the licking phase, as mice need to traverse to the reward spout after their decision. We show that in our configuration the time between touch and lick is about 300-500 ms (Fig. 4 b,c), allowing specific analyses of sensory-evoked neuronal activity that occurs before the lick events (Figs. 5b, h, 6d,e).

5. Behavioral and Technical Innovations

Compared to existing unrestrained tasks, our paradigm offers critical advances in regards to behavioral readouts and experimenter bias:

Time-Stamped Behavioral Data: Unlike freely moving tasks, for example a recent tactile discrimination task (Qi, Ye, Naskar, Inácio, & Lee, 2022), our approach integrates behavioral timestamping (such as whisker touches, lick events, whisker kinematics, head-direction, locomotion, position), enabling detailed alignment of behavioral parameters with neural activity.

Automatization: Our task design minimizes confounders through automated control, enabling the experimenter to leave the room, thereby enhancing reproducibility and robustness.

We hope these points better highlight the distinct contributions of our work and the opportunities they open for future research.

2/ Reversal learning behavior is not new. The observation that reversal learning can take more time than initial learning (not systematic) is widely reported across a large number of tasks both freely moving and head fixed. It is a widely used test of cognitive flexibility. See this review <https://www.ncbi.nlm.nih.gov/pmc/articles/PMC5018909/>. Reversal learning is not new for a tactile task either <https://www.nature.com/articles/s41586-020-2704-z>.

Response: We fully agree that reversal learning is not novel per se. However, our study does not claim novelty of reversal learning. Instead, we emphasize the integration of reversal learning within a tactile learning task for freely moving mice — an achievement that has not been demonstrated before, including the head-restrained study referenced by the reviewer (Banerjee et al., 2020). Furthermore, we show repeated reversal learning, allowing for an experimental setting in which mice continuously switch back and forth between rules. We demonstrate that this repeated rule reversal paradigm can be accommodated within an experimentally accessible time frame. For example, initial learning plus three subsequent rule reversals as shown in Fig. 2d were done in under one month and neuronal spike activity was monitored throughout.

3/ Recordings in freely moving behaving rodents during a tactile discrimination task have also been performed in the past. For example by Nikbaht et al. 10.1016/j.neuron.2018.01.003. This study shows that stimulus classification can be decoded from brain activity. Hence, decoding stimuli from brain activity of a freely moving animal is not new either. It has been widely done in head-fixed mice as well, the authors here do not bring any results that differs from previous knowledge.

Response: We fully agree that recordings in freely moving animals and stimulus decoding from neuronal activity are not novel per se. While our study does not claim novelty of these techniques, we demonstrate that these techniques can be implemented in a mouse task (the reviewer cited a rat task). We see this as a significant advance for the field and concur with reviewer 3 who points out “The establishment of this task in mice is of value to the scientific community because of the vast toolbox available to unravel the neuronal circuit basis for sensory perception,...”.

4/ Electrophysiological recordings in freely behaving animals are common. Here is a 2009 review by the Brecht lab 10.1016/j.conb.2009.08.005 about even more challenging approaches to do electrophysiology in freely behaving rodents. The miniscope used by the authors is not novel all. It has been used in 100s of studies in different behavioral context.

Response: We fully agree that electrophysiological and miniscope recordings in freely behaving animals are not novel per se. However, our study does not claim novelty in the techniques themselves. Instead, we emphasize the unique integration of these techniques within a tactile learning task for freely moving mice—an achievement that has not been demonstrated before (including the reviewer’s referenced work by Michael Brecht’s lab). By bridging the gap between head-fixed and freely moving paradigms, we provide a new framework for studying the

complexity of animal behaviors in a naturalistic context, with implications for studying sensory processing and neuronal mechanisms of learning in more naturalistic settings.

Limits of the analysis.

1/ There is no assessment of whether the task is barrel cortex dependent. This is a major issue as some tasks are and other not (Hong et al, 2018; Rodgers et al. 2021; Harrell et al. 2021). The value of the decoding analysis is diminished by the lack of casual evaluation.

Response: In regards to the barrel-cortex dependency of the task, we devised new sets of experiments. We physically ablated the barrel cortex bilaterally, similarly as in (Hong, Lacefield, Rodgers, & Bruno, 2018). Sham lesions served as control cohorts. In the first experiment, we addressed the cortex-dependency of task execution (pertaining to the reviewer's question). Barrel cortex ablations and sham operations were done in trained animals. Ablated animals, but not sham controls dropped below the performance threshold, demonstrating that task execution is dependent on barrel cortex (new Figs. 4c, S4).

Furthermore, in a second experiment, we performed barrel cortex ablation and sham operations in naive animals and trained them on the basic task rule. While sham controls showed normal learning times (~500 trials to reach expert level), the learning in barrel-cortex ablated animals was severely impaired, with a roughly four-fold increase in trials required for expert level performance. Furthermore, the performance of barrel cortex ablated animals remained marginally above expert-level performance and never reached similar d' levels as sham animals (new Figs. 4d,e, S4).

Together, these results show that both learning and task execution of the freely moving discrimination paradigm are dependent on the barrel cortex. The results are integrated into the revised manuscript (Figs. 4, S4).

In regards to the second point, we demonstrate that the decoding is causally dependent on intact whiskers (Fig. 4i). Here, the decoding accuracy significantly decreased when comparing lidocaine to neutral ointment sessions.

2/ The observation that you can decode gap aperture from whisker movement even with lidocaine in the pad is trivial. It is also trivial that one can decode from the tactile system. It would have been interesting to evidence the coding schemes at play i.e. what whisker movement triggers which type of activity.

Response: We addressed the question of which whisker movements trigger which type of activity by several new sets of analyses:

Locomotion tuning (new Fig. 5k), whisking (new Fig. S5), whisker angle (new Fig. 6a, b) and phase (new Fig. 6 c-e), head angle (new Fig. 6f) and spatial tuning (new Figs. 5i, j). A further analysis of specific discharge patterns and coding schemes was elaborated in another manuscript, which is now available as a preprint: <https://doi.org/10.1101/2025.03.11.642368>

3/ There is no use of the imaging data and no comparison between decoding with electrophysiology and imaging.

Response: We added new analyses of the imaging data, including a comparison between imaging and electrophysiological results (new Fig. 3f, g). In addition, we added a convolutional neural network (CNN) to classify neuronal activity from somatic calcium transients over expert sessions (new Fig. 3b, c). For the POM recordings, the CNN reached a comparable aperture decoding accuracy (87%) as compared to the decoding of aperture width from unit spike trains.

4/ There is no analysis of the evolution of neural activity with learning (has been done in the past in tactile tasks with precise measures under head fixation).

Response: We appreciate the reviewer's suggestion regarding the analysis of neural activity evolution during learning. We indeed find that neuronal activity in the somatosensory system undergoes profound changes as a function of learning. Most notably, we found that neuronal bursting evolves with learning. Specifically, burst-coder neurons (BCNs) emerge in cortical, thalamic, and extrathalamic regions, encoding task-relevant stimulus-outcome associations rather than physical stimulus identity. The number of BCNs increases with task proficiency, and their burst patterns dynamically adapt to rule switches, supporting the role of bursts as context-dependent teaching signals. In order to adequately present the content of the current study as well as the findings pertaining to the evolution of neural activity with learning, we believe that two parallel manuscripts are needed. We would therefore like to refer to our parallel study that focuses on the neural dynamics during learning: <https://doi.org/10.1101/2025.03.11.642368>

Reviewer #2 (Remarks on code availability):

Data and code are available on GitHub not on a proper repository

Response: The data and code are available on both Github and as a persistent repository on Zenodo (<https://doi.org/10.5281/zenodo.15051370>). We apologize that the Zenodo repository was only indicated as a reference in the Data and Code availability section. We also added the link now.

Reviewer #3 (Remarks to the Author):

The authors present a whisker-dependent sensory discrimination task in which freely moving mice are trained to distinguish aperture widths. The mice convey their percept by collecting condensed milk rewards at lick ports (on either end of a linear track) for some aperture widths while avoiding the lick ports for other widths to elude noise punishments. This task is adapted from a classic study (Krupa et al. 2004) in which freely moving rats were trained to discriminate apertures, and the main departure with respect to the current study is that the rats indicated the perceived aperture (narrow vs wide) by choosing the correct side in a two-port left/right apparatus. The establishment of this task in mice is of value to the scientific community because of the vast toolbox available to unravel the neuronal circuit basis for sensory perception, but the authors do not go as far as to take advantage of this toolbox. In its current form, the manuscript

reads more like a methods paper than a research article, so the authors will need to make significant improvements to publish this as such.

In the introduction, the central argument for why this task is an important development is that unlike the popular head-fixed paradigms for studying whisker-based touch sensation in mice, their freely moving paradigm allows "active exploration of the rodent's surroundings". This argument is under-developed. Even in head-fixed conditions, mice can and do move their whiskers which in many studies has been considered "active sensing" (see the work of the Kleinfeld, Ahissar, Peterson, and Feldman labs). In head-fixed virtual reality, rodents can also run and use their whiskers actively (see work from the Helmchen lab). Further, freely moving rodents can use their whiskers passively by holding them still and running along a wall or moving their head with fixed whiskers, so the authors need to be more precise about what they mean by active exploration. In my opinion, the head movements are the big difference in non-head-fixed conditions, so a more complete introduction needs to highlight the importance of head movements in whisker-based sensation and provide evidence from the literature to support (see work from the Diamond lab) and justify the precise advantages of their configuration.

Response: We appreciate the reviewer's feedback and have revised the manuscript to better articulate the advantages of our freely moving whisker-based sensory discrimination task. We concur with the reviewer, that the head movements are a significant advantage over restrained configurations and show in a new analysis how head-tracking can be used as a decision readout, in addition to commonly used lick analysis (see response to next comment). We also revised the introduction as suggested.

Additionally, we highlight another concrete advantage of our paradigm: the temporal separation between sensory sampling and licking, which minimizes confounding influences of lick-related activity during perception. In head-fixed paradigms, anticipatory licking often overlaps with sensory processing, making it challenging to disentangle sensory-evoked activity from lick-associated neural responses (for recent examples (Moberg et al., 2025; Petty & Bruno, 2024)). Our design mitigates this issue by ensuring that licking occurs only after a decision is made. We emphasized these points at several places through the manuscript (introduction, results and discussion).

The characterization of the task performance and aperture sensing capabilities of the mice (Figs 1-3) is adequate and the authors show that it depends on intact whiskers at the periphery and not on visual cues.

The analysis of the adaptation of motor behavior during learning (Fig. 4) is insufficient. Panels A, B, and C all focus on licking. In head-fixed go-nogo paradigms, indeed licking is the only "decision" variable, but here, it is apparent from their supplementary videos that in their freely moving configuration, advancing through the aperture towards the lick port is the first decision-revealing movement. The mouse never crosses the gap between the no-go aperture and the lick port in the Correct Rejection trial video and it approaches the lick port and licks in

the Hit trial video. They need to characterize this approach/stopping behavior, including head movement and orientations, as well as the whisker/head movements used to sample the aperture throughout training. As eluded to in the commentary above for the introduction, these are the elements of the behavior that are unique in their configuration compared to head-fixed go-nogo and will be of interest to the field. The only non-licking motor behavior presented is running along the track, which reflects task engagement but not the skill of performing the task.

Response: Thank you for these suggestions and for encouraging us to take advantage of the unrestrained animal behavior. Indeed, new analyses show that the approach/stopping behavior can be used as a readout of learning and as a proxy for the time-line of decision and task execution. To do so, we tracked the head trajectory in naive and expert mice during Hit and Correct Rejection (CR) trials. During CR, mice show a turning behavior which we then used to estimate the reaction time (the time between aperture touch and turning behavior) and reaction distance (how close the mouse approaches the lick port before turning around). Both the reaction time and reaction distance closely correlate with the performance, such that expert animals show the shortest reaction times and shortest reaction distances. This demonstrates that indeed decision variables can be obtained from non-licking behavior in an unrestrained paradigm. These results have been integrated into the revised manuscript (new Fig. 5a-c).

The electrophysiological recordings and imaging data are not adequately analyzed (Figs 5-6). Indeed, it is important to be able to carry out these measurements during the task performance, and it is impressive to record from many regions simultaneously, but classifying responsive percentages of the populations in their respective regions alone is not enough. Here are questions that can and should be answered:

1.) How is aperture width encoded by single cells and the population? Are there selective single cells? Is this different across the different brain regions?

Response: We appreciate and fully share the reviewers interest in using the presented paradigm to study the encoding of aperture cues across different brain regions and learning. We can confirm that aperture width is encoded by single neurons as well as by populations, there are selective single neurons and there are differences across brain regions and a strong relationship between aperture encoding and learning. In order to adequately present the content of the current study as well as the findings pertaining to coding and learning, we believe that two parallel manuscripts are needed. We would therefore like to refer to our parallel study that focuses on the coding related questions: <https://doi.org/10.1101/2025.03.11.642368>

2.) What other task-related parameters (licking, head movements, stopping, running, whisking, bilateral touch, etc.) are encoded and is this different in the different regions?

Response: As suggested, we now examined locomotion tuning (new Fig. 5k), whisking (new Fig. S5), whisker angle (new Fig. 6a, b), whisker phase (new Fig. 6c-e), head angle (new Fig. 6f), and spatial tuning (new Fig. 5i, j). These analyses reveal distinct neural representations of

movement and sensory variables, providing insights into how different regions contribute to task execution and sensory processing.

3.) For the classifier analysis, it is not clear what is happening in the 400 ms window chosen (400 ms after the trigger is crossed). Are the mice only whisking at the aperture during this window or are they already advancing to the reward? What about the temporal dynamics of the classifier performance? How does classification accuracy build up and does it do so with a certain relationship to decision? This can of course be checked in the Go trials only with decision being the time that the mouse advances its head beyond the aperture to collect reward.

Response: We appreciate the reviewer's request for clarification regarding the 400 ms analysis window. During this period, mice continue advancing toward the lick spout without pausing (see figure below), and stimulus sampling occurs concurrently with movement (approach or retraction). The 400 ms window was chosen to exclude licking-related activity while capturing sufficient sensory-related activity (see lick latencies: median 475 ms [IQR: 379–616 ms] during initial learning).

To examine the temporal dynamics of classifier performance, we now provide a time-resolved decoding analysis (new Fig. 4g). This analysis shows that decoding accuracy increases sharply after whisker-aperture touch, peaking at ~100 ms—well before the first lick. This suggests that the decision to lick or withdraw is computed within milliseconds, preceding overt behavioral responses (see retraction times in new Fig. 5b).

Beyond what is reported, I think there are two important dimensions that should be investigated to make the impact of the report sufficient for publication in Nature Communications as a research article:

1.) Does task performance require barrel cortex? Lesions and/or muscimol experiments could be designed or optogenetics if possible.

Response: We devised new sets of experiments in which we physically ablated the barrel cortex bilaterally, similarly as in (Hong et al., 2018). Sham lesions served as control cohorts. In the first experiment, we addressed the cortex-dependency of task execution (pertaining to the reviewer's question). Barrel cortex ablations and sham operations were done in trained animals. Ablated animals, but not sham controls dropped below the performance threshold, demonstrating that task execution is dependent on barrel cortex.

Furthermore, in a second experiment, we performed barrel cortex ablation and sham operation in naive animals and trained them on the basic task rule. While sham controls showed normal learning times (~500 trials to reach expert level), the learning in barrel-cortex ablated animals was severely impaired, with a roughly four-fold increase in trials required for expert level performance. Furthermore, the performance of barrel cortex ablated animals remained marginally above expert-level performance and never reached similar d' levels as sham animals.

Together, these results show that both learning and task execution of the freely moving discrimination paradigm are dependent on the barrel cortex. The results are integrated into the revised manuscript (Figs. 4, S4).

2.) Is bilateral whisker sensing required? The authors could perform the whisker plucking or lidocaine treatments to one side and see how this affects task performance and whisking/head movement strategy. If bilateral whisker sensing is needed, discuss how and where could these bilateral signals could be integrated.

Response: Thank you for bringing up this interesting question. We addressed the requirement of bilateral whisker sensing in two new experiments. Firstly, we trained an additional cohort to expert-level performance and then removed one aperture wing on each side of the linear track. All mice dropped under the expert-level threshold and discrimination was poor (new Fig. S4c). Secondly, as suggested, we trained two new cohorts to expert-level performance on 20 mm and 6 mm contrasts and then plucked the whiskers on one side. The performance dropped in all mice, however, only for the 6mm contrast the performance dropped under the expert-level threshold. We conclude that the task becomes more and more dependent on bilateral whisker sensing as the contrasts become small. For larger contrasts, when only one set of whiskers is available, mice may use additional facial parts such as the nose and cheek for discrimination. The results are integrated into the revised manuscript (new Fig. 4b, S4c).

Reviewer #3 (Remarks on code availability):

I did not assess the code.

References

- Ahissar, E., & Knutsen, P. M. (2008). Object localization with whiskers. *Biological Cybernetics*, 98(6), 449–458. <https://doi.org/10.1007/s00422-008-0214-4>
- Banerjee, A., Parente, G., Teutsch, J., Lewis, C., Voigt, F. F., & Helmchen, F. (2020). Value-guided remapping of sensory cortex by lateral orbitofrontal cortex. *Nature*, 585(7824), 245–250. <https://doi.org/10.1038/s41586-020-2704-z>
- Datta, S. R., Anderson, D. J., Branson, K., Perona, P., & Leifer, A. (2019). Computational neuroethology: A call to action. *Neuron*, 104(1), 11–24. <https://doi.org/10.1016/j.neuron.2019.09.038>
- Hong, Y. K., Lacefield, C. O., Rodgers, C. C., & Bruno, R. M. (2018). Sensation, movement and learning in the absence of barrel cortex. *Nature*, 561(7724), 542–546. <https://doi.org/10.1038/s41586-018-0527-y>
- Krakauer, J. W., Ghazanfar, A. A., Gomez-Marin, A., MacIver, M. A., & Poeppel, D. (2017).

Neuroscience needs behavior: Correcting a reductionist bias. *Neuron*, 93(3), 480–490.

<https://doi.org/10.1016/j.neuron.2016.12.041>

Krupa, D. J., Matell, M. S., Brisben, A. J., Oliveira, L. M., & Nicolelis, M. A. (2001). Behavioral properties of the trigeminal somatosensory system in rats performing whisker-dependent tactile discriminations. *The Journal of Neuroscience: The Official Journal of the Society for Neuroscience*, 21(15), 5752–5763. <https://doi.org/10.1523/jneurosci.21-15-05752.2001>

Krupa, D. J., Wiest, M. C., Shuler, M. G., Laubach, M., & Nicolelis, M. A. L. (2004).

Layer-specific somatosensory cortical activation during active tactile discrimination. *Science (New York, N.Y.)*, 304(5679), 1989–1992. <https://doi.org/10.1126/science.1093318>

Miller, C. T., Gire, D., Hoke, K., Huk, A. C., Kelley, D., Leopold, D. A., ... Niell, C. M. (2022).

Natural behavior is the language of the brain. *Current Biology: CB*, 32(10), R482–R493.

<https://doi.org/10.1016/j.cub.2022.03.031>

Mitchinson, B., Grant, R. A., Arkley, K., Rankov, V., Perkon, I., & Prescott, T. J. (2011). Active vibrissal sensing in rodents and marsupials. *Philosophical Transactions of the Royal Society of London. Series B, Biological Sciences*, 366(1581), 3037–3048.

<https://doi.org/10.1098/rstb.2011.0156>

Moberg, S., Garibbo, M., Mazo, C., Gilad, A., Schmitz, D., Costa, R. P., ... Takahashi, N. (2025).

Distinct roles of cortical layer 5 subtypes in associative learning (p. 2025.01.07.631500).

<https://doi.org/10.1101/2025.01.07.631500>

Nikbakht, N., Tafreshiha, A., Zoccolan, D., & Diamond, M. E. (2018). Supralinear and

supramodal integration of visual and tactile signals in rats: Psychophysics and neuronal mechanisms. *Neuron*, 97(3), 626–639.e8. <https://doi.org/10.1016/j.neuron.2018.01.003>

Oram, T. B., Tenzer, A., Saraf-Sinik, I., Yizhar, O., & Ahissar, E. (2024). Co-coding of head and whisker movements by both VPM and POm thalamic neurons. *Nature Communications*, 15(1), 5883. <https://doi.org/10.1038/s41467-024-50039-z>

Petty, G. H., & Bruno, R. M. (2024). Attentional modulation of secondary somatosensory and

visual thalamus of mice. <https://doi.org/10.7554/elife.97188.1>

Qi, J., Ye, C., Naskar, S., Inácio, A. R., & Lee, S. (2022). Posteromedial thalamic nucleus activity significantly contributes to perceptual discrimination. *PLoS Biology*, *20*(11), e3001896. <https://doi.org/10.1371/journal.pbio.3001896>

Rosenberg, M., Zhang, T., Perona, P., & Meister, M. (2021). Mice in a labyrinth show rapid learning, sudden insight, and efficient exploration. *eLife*, *10*. <https://doi.org/10.7554/eLife.66175>

Sachdev, R. N. S., Berg, R. W., Champney, G., Kleinfeld, D., & Ebner, F. F. (2003). Unilateral vibrissa contact: changes in amplitude but not timing of rhythmic whisking. *Somatosensory & Motor Research*, *20*(2), 163–169. <https://doi.org/10.1080/08990220311000405208>

Sumser, A., Isaías-Camacho, E. U., Mease, R. A., & Groh, A. (2024). Differential representation of active and passive touch in mouse somatosensory thalamus (p. 2024.07.16.603697). <https://doi.org/10.1101/2024.07.16.603697>

Voigts, J., Sakmann, B., & Celikel, T. (2008). Unsupervised whisker tracking in unrestrained behaving animals. *Journal of Neurophysiology*, *100*(1), 504–515. <https://doi.org/10.1152/jn.00012.2008>

Reviewer #1 (Remarks to the Author):

The authors addressed all my points adequately. I have no additional comments. Congratulations on an impressive paper!

Response: Many thanks!

Reviewer #2 (Remarks to the Author):

My main point from the previous round of review was that there is neither technical nor biological novelty in the results reported in the study. The technical breakthrough, if any, is mainly to have successfully put different established techniques together.

Response: As outlined in our previous response and presented in the manuscript and supported by other reviewers, this study introduces a technologically novel and biologically informative platform, demonstrating multi-site neural recordings during a tactile discrimination learning task in freely moving mice. This has not been achieved before and presents an important technological breakthrough over the current head-fixed paradigms. Our study goes far beyond the combination of existing techniques by providing "an enabling platform" that allows for discoveries previously inaccessible in head-fixed settings (see also comments below).

This is certainly a lot of carefully done work but, in the end, what is really the added value for the reader beyond the fact that putting these techniques together is feasible ?

Response: The reviewer acknowledges the technical soundness of our study. Demonstration of feasibility is an inherent step in technological advance, including for this novel approach and is needed to make it applicable by the scientific community. We added a new section in which we directly discuss the applicability of the platform: "The setup is robust, modular, and largely automated. Once configured, experiments can run without the experimenter in the room, reducing potential biases. Maintenance is minimal aside from routine cleaning after the experiments. All trained mice in our hands learned the task successfully, and detailed documentation ¹ makes the system readily adoptable. The modularity of the hardware and acquisition software ² allow easy customization to other research approaches, including integration with silicon probes, Neuropixels, optogenetics, or custom imaging setups.

This versatility enables the platform to address a wide range of research questions. For example it can be used to study spatial navigation (Fig. 5i,j) and head-orientation (Fig. 6f), or be adapted into a delay-based task to probe short-term memory mechanisms ^{3,4} by increasing the distance between the aperture and the lick spout. The paradigm also allows investigation of perceptual uncertainty by training mice at near-threshold contrasts (Fig. 2f), extending work on sensory discrimination under ambiguous conditions ⁵⁻⁸. Furthermore, it is suitable for probing how internal

states such as sleep, hunger⁹, or social experience influence learning, as well as for examining the contribution of specific neural circuits – such as the higher-order thalamic nucleus POm (Fig. 3) – to task performance^{10–12}. “

Moreover, our lesion experiments, whisker manipulations, and multi-region electrophysiology provide new insights into how tactile decisions are distributed across brain areas, and how learning shapes these representations—advances that go far beyond a technical proof-of-concept.

The authors argue that it is the first time this type of behavioral task was successfully implemented in freely moving mice. I agree with them. Yet I maintain my assessment that this is a nice result but a quite specialized result and not necessarily at the level of novelty expected for a Nature Communications paper. Many tasks published in this journal are novel but are accompanied with novel biological results, or a new technique.

Response: We believe that decisions about the scope and editorial fit of the studies are best left to the editors. We would like to refer to a few examples of excellent methods papers, including novel mouse tasks published in Nature Communications, which do not contain biological insights^{2,13–17}.

In addition, our manuscript goes beyond a methods paper. For instance, we use the novel paradigm to demonstrate in a freely-moving setup:

- the necessity of the barrel cortex and bilateral whisker input for efficient task learning and execution and new analysis of evolving aperture selectivity in barrel cortex (Fig. 4),
- modulation of thalamocortical neurons not only by diverse behavioral states but also by extrinsic parameters (spatial- and head-tuned units) (Figs. 5, 6),
- the decodable emergence of decision variables from motor behavior (Fig. 5a-c).
- the capability of the olfactory system alone to drive texture discrimination (Fig. S4a) in freely-moving paradigms thought to be purely tactile and that our paradigm mitigates these limitations.

These findings go beyond feasibility and provide concrete biological insights that can serve as reference points for future work in systems neuroscience.

The author argue that their task is better documented than in the past or that it integrates multiple recording modalities. These important aspects are expected in high quality studies reporting new discoveries nowadays and do not replace a biological result.

Response: While we agree that high-quality documentation and multimodal integration should be standard, we respectfully note that they remain under-realized goals in many domains. In this work, we not only integrate modalities (electrophysiology, imaging, behavior tracking), but make the tools and datasets accessible and reproducible, exceeding typical standards. In order to facilitate adopting the platform in other labs, we included detailed parts lists of the platform

modules, 3D blender and CAD files, images, analyses scripts to extract behavioral measures and data <https://doi.org/10.5281/zenodo.15051370>

This positions our work as a practical and scalable resource for others studying various research questions including neuronal mechanisms of learning and behavior in freely moving animals and supports the trend to study brain functions in ecologically realistic settings.

The authors must be commended for the additional data provided, in particular for lesion data. They clearly demonstrate that barrel cortex eases task learning and that the task is still feasible without barrel cortex. This is important data for future users and for understanding the specific role of barrel cortex in task performance. It also shows that this is just a sensory discrimination task and that it can be solved even if not perfectly without cortical sensory processing. Hence cortex is partially dispensable/necessary. This should be reported in the abstract.

Response: We thank the reviewer for highlighting the importance of the lesion data. We have now updated the abstract to reflect this key finding: that the barrel cortex is essential for normal learning speed and peak performance, but that task learning can still occur—albeit at reduced efficiency—without it. From an evolutionary perspective, this important result suggests that learning per se can be achieved without cortical involvement, albeit at much lower pace and the evolution of the neocortex may be viewed as an optimization process to allow faster learning (Fig. 4d) and better performance (Fig. 4e).

The authors claim in their abstract that this task is a platform to study cognitive aspects. There is in fact not much in the results corroborating this view. The task is essentially sensory, only half cortex dependent, and the reversal learning, even when repeated, does not accelerate learning in comparison with the initial learning. So, there is really nothing cognitive in this task. This should be removed from the abstract.

We respectfully note that the reviewer's concern about cognitive aspects was not raised during the initial round of review, even though references to cognitive processes were already present in the original submission.

Our learning paradigm fundamentally engages well-established cognitive functions, including associative learning, reinforcement learning, decision-making, and cognitive flexibility. For instance, reversal learning (Figs. 2c–e, S1, S2) — widely used as a model for studying cognitive flexibility^{18,19} — is a core feature of the task.

Specifically, our platform supports investigations of cognitive processes such as:

- Associative learning and memory: Mice learn to associate stimulus features with outcomes (reward or punishment) over repeated trials. This associative learning is impaired after cortical lesions (Fig. 4c–e), highlighting its dependence on neural circuits beyond simple sensory processing.

- Rule reversal learning: Mice successfully adapt to reversed task rules, demonstrating behavioral flexibility—a hallmark of cognitive flexibility (Figs. 2c–e, S1, S2).
- Generalization and abstraction: Across repeated reversals, learning becomes more efficient (Figs. 2d, S1f), suggesting that mice abstract and generalize task structure beyond simple stimulus-response mappings.
- Insight-like learning: Performance improvements occur abruptly after periods of exploration, consistent with insight-like learning phenomena (Figs. 2b, S1d).

These features go beyond sensory detection and reflect cognitive processes such as memory, abstraction, and strategy updating.

Given that the reviewer's definition of cognitive processes appears to differ from the broader view widely accepted in neuroscience, we have nonetheless toned down the reference to cognitive aspects in the abstract to accommodate this feedback, while maintaining scientific accuracy.

The authors have more thoroughly analyzed the representation of touch, in multiple regions of the tactile system, yet they provide no evidence of a particular progression of these representations that could explain why barrel cortex is needed to learn faster the task. The results are not mentioned in the abstract showing their lack of scientific relevance.

Response: We added the results of the lesion results to the abstract. In addition, we added a new analysis of the longitudinal progression of aperture selectivity in barrel cortex units, which provides an explanation for why barrel cortex is needed to learn the task faster and more efficiently (Fig. 4f and abstract).

Finally, we added a new discussion of the utility of the platform to investigate the neuronal representations of task-reward contingencies across learning in the thalamocortical system: "For instance, in a parallel study we successfully utilized the learning platform to monitor neuronal dynamics over multiple learning and reversal stages to investigate neuronal encoding of reward contingencies during learning ¹².

The observation of sensitivity to diverse behavioral parameters is important although not surprising given the large amount of literature in the visual (Carandini lab, Striker lab, and many other) or auditory (Kuchibhotla lab, Froemke lab, King lab) system describing this. This is of course the first time this is extensively documented in a freely moving tactile task in mice. Maybe an important confirmation but not surprising in any way.

Response: We added further constructive discussions and cited additional relevant literature throughout the revised discussion.

Overall, I maintain my statement. A solid work with very little novelty.

Response: The conclusion of little novelty is not supported by the results we present in the manuscript.

Reviewer #3 (Remarks to the Author):

I would like to commend the authors for having addressed all of the comments. They have done a thorough rebuttal and greatly enhanced the scope and novelty of the study, even sometimes going beyond what was asked of them. The work is complete and suitable for publication.

Response: Many thanks!

I would suggest one last read through as there are some small typos and wording issues. One example in line 279 :

"The complete removal of whiskers, deteriorated the performance of expert animals below performance threshold (Fig. 4a). In contrast, animals continued to perform at expert level with IR backlights and IR overhead LEDs turned off, demonstrating that visual cues were not used to solve the task (Fig. S4b)."

This second sentence should explicitly indicate that those animals are whisker-intact. it reads as if those tests were done on the whisker-removed animals mentioned in the sentence before.

Response: Done, thank you for pointing this out. The sentence reads now: "In a separate experiment, animals with intact whiskers continued to perform at expert level with IR backlights and IR overhead LEDs turned off, demonstrating that visual cues were not used to solve the task (Fig. S4b)."

Reviewer #3 (Remarks on code availability):

I did not review the code.

1. Heimburg, F. *et al.* *Supplementary Materials for: "A Tactile Discrimination Task to Study Neuronal Dynamics in Freely-Moving Mice"* (2024). (Zenodo, 2024).
doi:10.5281/ZENODO.13369686.
2. Klumpp, M. *et al.* Syntalos: a software for precise synchronization of simultaneous multi-modal data acquisition and closed-loop interventions. *Nat. Commun.* **16**, 708 (2025).
3. Bolkan, S. S. *et al.* Thalamic projections sustain prefrontal activity during working memory maintenance. *Nat. Neurosci.* **20**, 987–996 (2017).
4. Schmitt, L. I. *et al.* Thalamic amplification of cortical connectivity sustains attentional control. *Nature* **545**, 219–223 (2017).
5. Ollerenshaw, D. R., Zheng, H. J. V., Millard, D. C., Wang, Q. & Stanley, G. B. The adaptive trade-off between detection and discrimination in cortical representations and behavior. *Neuron* **81**, 1152–1164 (2014).
6. Lesica, N. A. *et al.* Adaptation to stimulus contrast and correlations during natural visual stimulation. *Neuron* **55**, 479–491 (2007).
7. Musall, S. *et al.* Tactile frequency discrimination is enhanced by circumventing neocortical adaptation. *Nat. Neurosci.* **17**, 1567–1573 (2014).
8. Gerdjikov, T. V., Bergner, C. G., Stüttgen, M. C., Waiblinger, C. & Schwarz, C. Discrimination of vibrotactile stimuli in the rat whisker system: behavior and neurometrics. *Neuron* **65**, 530–540 (2010).
9. Padamsey, Z., Katsanevaki, D., Dupuy, N. & Rochefort, N. L. Neocortex saves energy by reducing coding precision during food scarcity. *Neuron* **110**, 280-296.e10 (2022).
10. La Terra, D. *et al.* The role of higher-order thalamus during learning and correct performance in goal-directed behavior. *Elife* **11**, (2022).

11. Qi, J., Ye, C., Naskar, S., Inácio, A. R. & Lee, S. Posteromedial thalamic nucleus activity significantly contributes to perceptual discrimination. *PLoS Biol.* **20**, e3001896 (2022).
12. Heimburg, F., Timm, J., Saluti, N. M. & Groh, A. Distributed burst activity in the thalamocortical system encodes reward contingencies during learning. *bioRxiv* 2025.03.11.642368 (2025) doi:10.1101/2025.03.11.642368.
13. Wickersham, I. R., Sullivan, H. A. & Seung, H. S. Axonal and subcellular labelling using modified rabies viral vectors. *Nat. Commun.* **4**, 2332 (2013).
14. Fernandez Lahore, R. G. *et al.* Calcium-permeable channelrhodopsins for the photocontrol of calcium signalling. *Nat. Commun.* **13**, 7844 (2022).
15. Bernal Sierra, Y. A. *et al.* Potassium channel-based optogenetic silencing. *Nat. Commun.* **9**, 4611 (2018).
16. Chaudhary, U. *et al.* Spelling interface using intracortical signals in a completely locked-in patient enabled via auditory neurofeedback training. *Nat. Commun.* **13**, 1236 (2022).
17. Aoki, R., Tsubota, T., Goya, Y. & Benucci, A. An automated platform for high-throughput mouse behavior and physiology with voluntary head-fixation. *Nat. Commun.* **8**, (2017).
18. Izquierdo, A., Brigman, J. L., Radke, A. K., Rudebeck, P. H. & Holmes, A. The neural basis of reversal learning: An updated perspective. *Neuroscience* **345**, 12–26 (2017).
19. Schoenbaum, G., Roesch, M. R., Stalnaker, T. A. & Takahashi, Y. K. A new perspective on the role of the orbitofrontal cortex in adaptive behaviour. *Nat. Rev. Neurosci.* **10**, 885–892 (2009).